# Diffusion-mediated HEI10 coarsening can explain meiotic crossover positioning in *Arabidopsis*

Chris Morgan 🆔 1,4, John A. Fozard 🆔 1,4, Matthew Hartley[1], Ian R. Henderson[2], Kirsten Bomblies[3] & Martin Howard 🆔 1✉

In most organisms, the number and distribution of crossovers that occur during meiosis are tightly controlled. All chromosomes must receive at least one 'obligatory crossover' and crossovers are prevented from occurring near one another by 'crossover interference'. However, the mechanistic basis of this phenomenon of crossover interference has remained mostly mysterious. Using quantitative super-resolution cytogenetics and mathematical modelling, we investigate crossover positioning in the *Arabidopsis thaliana* wild-type, an over-expressor of the conserved E3 ligase HEI10, and a *hei10* heterozygous line. We show that crossover positions can be explained by a predictive, diffusion-mediated coarsening model, in which large, approximately evenly-spaced HEI10 foci grow at the expense of smaller, closely-spaced clusters. We propose this coarsening process explains many aspects of *Arabidopsis* crossover positioning, including crossover interference. Consistent with this model, we also demonstrate that crossover positioning can be predictably modified in vivo simply by altering HEI10 dosage, with higher and lower dosage leading to weaker and stronger crossover interference, respectively. As HEI10 is a conserved member of the RING finger protein family that functions in the interference-sensitive pathway for crossover formation, we anticipate that similar mechanisms may regulate crossover positioning in diverse eukaryotes.

---

[1] John Innes Centre, Norwich Research Park, Norwich, UK. [2] Department of Plant Sciences, University of Cambridge, Cambridge, UK. [3] Institute of Molecular Plant Biology, Department of Biology, Swiss Federal Institute of Technology (ETH) Zürich, Zürich, Switzerland. [4]These authors contributed equally: Chris Morgan, John A. Fozard. ✉email: martin.howard@jic.ac.uk

 

A central feature of meiosis is the reciprocal exchange of DNA between pairs of homologous chromosomes via the formation of crossovers (COs)[1,2]. COs are crucial for promoting stable chromosome segregation during meiosis I and for introducing genetic diversity within offspring. COs are generated by the repair of programmed DNA double-strand breaks (DSBs) during meiotic prophase I, when chromosomes are organised along the meiotic axis into threadlike arrays of chromatin loops. In most species, DSBs vastly outnumber COs, with most DSBs being repaired via the formation of recombination intermediate (RI) joint molecules that are channelled into a non-crossover repair pathway[1]. The relative positions, and therefore numbers, of RIs that progress to form COs are tightly constrained: the formation of a CO at one chromosomal position inhibits the formation of additional COs nearby. This phenomenon, crossover interference (COI)[3,4], was first described over a century ago[5] and occurs widely across eukaryotes.

The mechanisms that underlie CO positioning, including COI, still remain to be fully elucidated, despite the proposal of elegant theoretical models[4]. In *Arabidopsis*, altered cellular abundance of HEI10, a conserved E3-ligase domain-containing protein, is known to alter CO numbers[6]. Over-expression of HEI10 has been shown genetically to weaken COI[7]. However, the effect of HEI10 dosage on the positioning of COs along prophase I chromosomes has remained cytologically unexplored. During meiotic prophase I, HEI10 localises as discrete foci along fully-synapsed pachytene chromosomes, initially in numbers that correlate with MSH4-labelled RIs[8]. These foci undergo a dynamic reduction in number (>100 foci to <15 foci per cell) from early to late-pachytene, with late-pachytene foci co-localising with CO-designated sites marked by MLH1[8]. It is therefore tempting to hypothesise that the dynamic reduction in HEI10 foci, which occurs at the visible RI to CO transition, may directly regulate CO positioning.

Here, to test this idea, we use super-resolution microscopy to examine the behaviour of HEI10. Accordingly, we find HEI10 behaviour is consistent with a coarsening process, and we propose that the coarsening dynamics of HEI10 is itself the primary driver of CO positioning and COI in *Arabidopsis*. To elucidate the mechanistic basis of HEI10 coarsening, we then take a modelling approach, comparing computational simulations of HEI10 coarsening dynamics with 3D-SIM measurements of HEI10 intensity in fixed pachytene cells. We find that our simulations are capable of recapitulating and predicting a variety of experimental observations in both wild-type plants and in plants with variable HEI10 dosage. Hence, we propose that HEI10-controlled CO positioning and COI can be explained by a diffusion-mediated coarsening process.

## Results

**HEI10 patterning in wild-type *Arabidopsis*.** To quantify HEI10 focus intensity and distribution, we developed a method for measuring HEI10 immunofluorescence intensity along individually segmented synaptonemal complexes (SCs) (stained by ZYP1) of fixed pachytene bivalents, imaged using 3D-SIM microscopy (Fig. 1A, Supplementary Fig. 1). This method revealed a widely varying HEI10 intensity as a function of position in μm along the SC, with numerous tightly localised peaks of fluorescence that could be identified as foci (see 'Methods' for precise definition). Due to differences in the relative compaction of chromatin along prophase I chromosomes[9], the optimal metric for measuring COI is μm of prophase I chromosome length rather than, for example, Mb of DNA[3]. In late-pachytene cells, we cytologically mapped CO-designated sites (late-HEI10 foci) in units of SC length by identifying the positions of large, discrete intensity peaks. We detected an average of 11.0 (s.d. ± 2.9) late-HEI10 foci per cell (n

= 104 cells) using this automated approach, which is comparable with previous values obtained from manual counts[8], and an average total SC length per cell of 240μm (s.d. ± 32 μm, n = 203 cells). This total SC length is higher than previous values obtained from EM analysis (~165 μm)[10] and may result from this method's enhanced ability to detect length excursions along the z-axis, or from differences in slide preparation methods. Using this technique, we also measured the total intensity of HEI10 at all stages (Fig. 1B, left), and of late-stage HEI10 specifically in foci (Fig. 1B, middle left), along individual bivalents, as normalized by total HEI10 bivalent intensity in the cell and total focal HEI10 intensity in the cell, respectively. The rationale for using per-cell normalized intensity values, as opposed to absolute intensity values, is that this approach controls for the influence of cell-to-cell variation in factors such as antibody binding and differences in the local slide environment, that can cause large variations in absolute intensity between cells (even between cells on the same slide). Comparing normalized intensity values also allowed us to modify imaging conditions for each cell to optimise the quality of 3D-SIM imaging, avoid the introduction of undesirable reconstruction artefacts, and also controls for any small differences in staging, as all foci within the same cell will necessarily be at the same stage. We found these per-cell normalized HEI10 intensity measures for mid- and late-stage cells positively correlate with SC length as normalized by total SC length within the cell. This correlation held also within a subset of the data in which we only considered bivalents with two late HEI10 foci, thereby reducing any influence of focus number on total focal HEI10 intensity (Fig. 1B, middle right). The normalized intensity of individual late HEI10 foci was also negatively correlated with the number of foci per bivalent: foci on bivalents with only a single late-HEI10 focus were brighter than foci on bivalents with two late-HEI10 foci, which in turn were brighter than foci on bivalents with three late-HEI10 foci (Fig. 1B, right). To further support this finding, we carried out additional analysis on both pseudo-widefield and 3D-SIM images, generated from the same raw imaging data, from a sub-sample of 10 wild-type late-pachytene cells using an alternative image-analysis approach (see Methods) and identified the same, significant negative relationship between late-HEI10 focus relative intensity and late-HEI10 focus number per bivalent (Supplementary Fig. 2). Taken together, these results support loading of HEI10 along pachytene bivalents in an SC length-dependent manner (with longer bivalents loading more HEI10), with an approximately fixed amount of HEI10 loaded per μm of SC, both at and between RIs. The subsequent HEI10 behaviour is then consistent with a coarsening process, where a fixed pool of HEI10 per bivalent first clusters into small foci at RIs, and then subsequently into larger foci, which grow at the expense of smaller ones (Fig. 1C). By late-pachytene, the HEI10 distribution could further coarsen to generate large, bright foci at CO-designated sites.

**Modelling diffusion-mediated HEI10 coarsening.** How such coarsening might arise mechanistically to generate a small number of approximately evenly-spaced HEI10 foci was not immediately clear. Due to the experimental limitations of super-resolution live-cell imaging over extended time periods, it is currently unfeasible to probe HEI10 coarsening behaviour in real-time. To elucidate the mechanistic basis of HEI10 coarsening, we, therefore, took a modelling approach, comparing simulations of HEI10 coarsening dynamics with 3D-SIM measurements of HEI10 intensity in fixed pachytene cells.

The principles underlying the coarsening model are summarised in Fig. 2a, b. In brief, a synapsed pachytene bivalent is considered as a one-dimensional structure, with the simulated SC

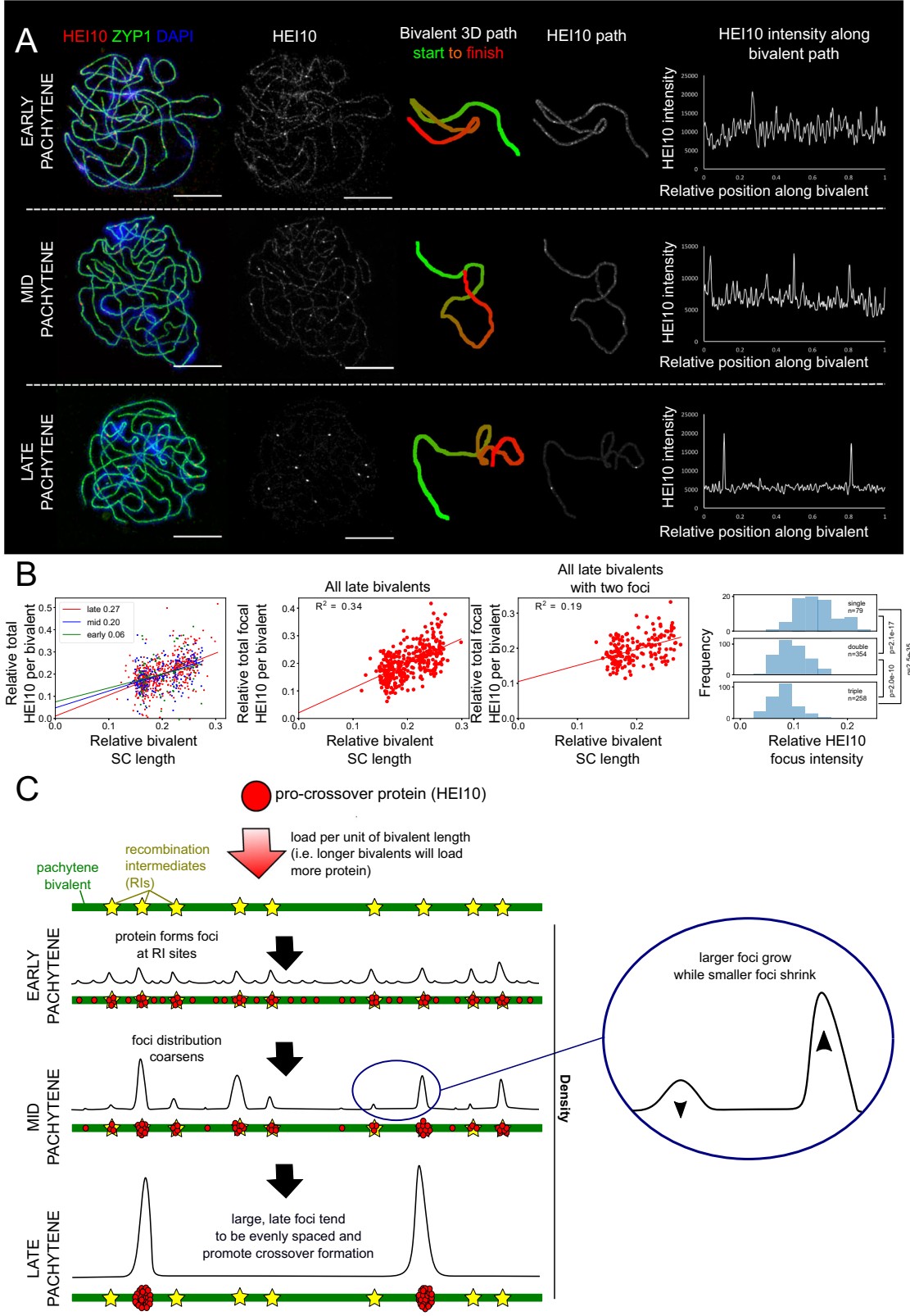

length drawn at random from the experimental bivalent SC length distributions. RIs were placed at random positions along the five bivalents, in accordance with the cytologically defined random or weakly-interfering nature of early recombination nodules in plants[11]. The total number of RIs on the five bivalents was consistent with cytologically reported *A. thaliana* RI values[12] and early pachytene HEI10 foci number[8], and for each bivalent

was also proportional to SC length. HEI10 is able to diffuse along the SC, in agreement with the behaviour of the HEI10 homologue, ZHP-3, in *C. elegans*[13]. At each RI, HEI10 is absorbed from the SC into immobile foci at a constant rate $\alpha$ and escapes back onto the SC at a rate $\beta$. The rate $\beta$ itself depends non-linearly on the amount of HEI10 in the focus, where larger foci have reduced escape rates. In this way, larger foci tend to

**Fig. 1 HEI10 protein dynamics in pachytene nuclei. A** HEI10 immuno-fluorescent intensity, focus number and position quantitatively measured along fully-synapsed bivalents in early, mid and late-pachytene nuclei stained for HEI10 (red), ZYP1 (green) and DAPI (blue), and imaged using 3D-SIM (images shown: maximum intensity projections of 3D image stacks). Representative images are shown. Data were collected from 75 early, 205 mid and 520 late pachytene bivalents for further analysis. Scale bars = 5 μm. **B** Left: total HEI10 intensity per bivalent, relative to total HEI10 bivalent intensity in the cell, plotted against bivalent SC length, relative to total measured SC length in the cell. Data (dots) coloured by pachytene stage (data from 75/205/520 bivalents for early/mid/late stages). Numbers in legend indicate R-squared values for linear regression. Middle left: same as left, but for HEI10 specifically in foci for late-stage cells, relative to sum of all HEI10 foci intensities in the cell (data from 370 bivalents). Middle right: same as middle left, but for late-stage cells with two HEI10 foci (data from 177 bivalents). Lines: linear regression best-fits (least-squares); colour-coded by stage. Right: distributions of HEI10 focus intensity for cases with 1, 2, 3 late HEI10 foci relative to sum of all HEI10 foci intensities in the cell. Numbers on right show (Bonferroni corrected) $p$-values from a one-sided Dunn's test of multiple comparisons using rank sums, following a Kruskal–Wallis test ($\chi^2 = 160.3, df = 2, p < 2.2 \times 10^{-16}$). Exact $p$-values are as follows: 1 vs 2 foci ($Z = 8.6, p$ unadjusted $= 7.1 \times 10^{-18}, p$ adjusted $= 2.1 \times 10^{-17}$), 1 vs 3 foci ($Z = 12.5, p$ unadjusted $= 8.4 \times 10^{-36}, p$ adjusted $= 2.5 \times 10^{-35}$), 2 vs 3 foci ($Z = 6.5, p$ unadjusted $= 6.6 \times 10^{-11}, p$ adjusted $= 2.0 \times 10^{-10}$). All pairwise comparisons are significant, with adjusted $p$-value < 0.0005. **C** From (**A**) HEI10 appears to show coarsening behaviour: HEI10 is distributed along the bivalents and at RIs (where it clusters into foci) at a fixed density (as suggested by data in **B**). As meiosis progresses, HEI10 distribution coarsens with larger foci growing at the expense of smaller foci. At late times, closely-spaced COs tend to be inhibited, reflecting COI. Source data are provided as a source data file.

accumulate HEI10, depleting the HEI10 substrate surrounding them, and therefore grow at the expense of smaller foci. As observed in related systems[14], such a decreasing escape rate can destabilize a uniform steady state, in this case with equal HEI10 at each focus. Foci can then undergo competitive coarsening towards a state in which one focus would contain the vast majority of the HEI10. If reached, the model predicts that a configuration with a single focus is then stable, as there is then nowhere else for HEI10 to accumulate, thereby naturally explaining the obligate CO.

Owing to the limited duration of the pachytene sub-stage of prophase I (up to 10 h[15] in *A. thaliana*), the coarsening process often does not resolve to a single focus. Instead, at the end of the 10 h simulation, the majority of HEI10 is contained in a small number of foci, which correspond to CO-designated sites. Initially in the simulation, HEI10 is distributed uniformly along the bivalents. However, at RIs, HEI10 is initially present at random amounts, drawn from a truncated normal distribution (Fig. 2b). To account for the preferential initiation of synapsis near telomeres[10,16], initial amounts of HEI10 at foci near bivalent ends in the model were greater than for those elsewhere (Fig. 2b, the spatially varying function *f*). This latter feature was, however, dispensable, albeit with decreased agreement between simulations and observations (Supplementary Fig. 3). Overall, the initial loading of HEI10 in the model is a fixed amount per μm of SC, consistent with cytological observations (Fig. 1B), meaning longer bivalents in the model possess a greater initial overall amount of HEI10 than shorter bivalents. We emphasise that the model is of only moderate complexity, with only five dynamical parameters and another six parameters specifying initial conditions beyond those specified by the experimental measurements of SC lengths and known pachytene duration, and is tightly constrained by our imaging data (see parameter values in Fig. 2b and Methods).

**Fitting the coarsening model to wild-type imaging data.** We next simultaneously fitted the model parameters to four aspects of cytologically generated *A. thaliana* meiotic datasets: late-HEI10 focus number, late-HEI10 focus positions, CO homoeostasis and dynamical timescale. Using our 3D-SIM late-pachytene data, late-HEI10 focus positions were obtained for all five SCs in each cell: the overall number and relative positions of late-HEI10 foci (in units of SC length) along each bivalent were extracted. As expected from COI, late-HEI10 focus number is tightly restricted along bivalents (Fig. 2c, left), with most bivalents experiencing only one, two or three late-HEI10 foci, with no zero late-HEI10 focus bivalents (hence, as expected, CO assurance is functioning normally in this material). The late-HEI10 focus distribution also follows an

expected pattern, with unimodal, bimodal and trimodal distributions being observed for single-, double- and triple- late-HEI10 focus bivalents, respectively (Fig. 2d–h, left). Using our model, we obtained closely matched distributions for both late-HEI10 focus number and positioning (Fig. 2c–h, right). CO homoeostasis describes the phenomenon where CO number shows only a weak linear dependence on DSB number[17]. In Arabidopsis, it has previously been shown by Xue et al. that reducing meiotic DSB numbers causes a smaller, but significant, reduction in CO numbers[18], with the authors commenting that this relationship is directly proportional when the single obligate CO is subtracted for each chromosome. We found that when we reduced the density of RIs in our model (with otherwise default parameters, including wild-type SC lengths) to match the previously published reductions in DSB number (~30% and ~40% reduction), we were able to fit model parameters to the expected reduction in CO number (~13% and ~17% reduction[18], respectively) (Fig. 2i). We also found that we were able to fit model parameters such that the coarsening dynamics of our model occurred within the experimentally observed ~10 h timeframe[15]. Smaller HEI10 foci could endure for several hours in our simulations, consistent with their frequent cytological observation (Fig. 2j).

**Testing model predictions against wild-type imaging data.** Once optimum model parameters were defined, we then further validated our model by comparing model predictions against cytologically generated data that the model was not fitted to. Model simulations indicated that, on double-CO bivalents, for an arbitrarily aligned SC, the intensity of the left late-HEI10 focus generally increased as the position of the midpoint between the two foci (as measured from the left end) also increased (Fig. 2k, right). The biophysical explanation for this observation within our model is that, during the coarsening process, the recombination-intermediate with greater 'catchment area' for siphoning HEI10 molecules will tend to accumulate more HEI10 molecules within their foci. After analysing the cytological data, we indeed found such a positive correlation (Fig. 2k, left). Similar reasoning also explains the phenomenon of COI within our model: closely-spaced COs are inhibited as they lack sufficient catchment area for HEI10 addition and are therefore out-competed by spatially isolated foci surrounded by a larger catchment area. We found that the distribution of relative foci intensities for single-, double- and triple-CO bivalents generated from model simulations (Fig. 2l, right) also closely resembled the distributions of relative late-HEI10 focus intensities that were initially observed from experiments (Fig. 2l, left). Additionally, in female *A. thaliana* meiocytes, it has previously been

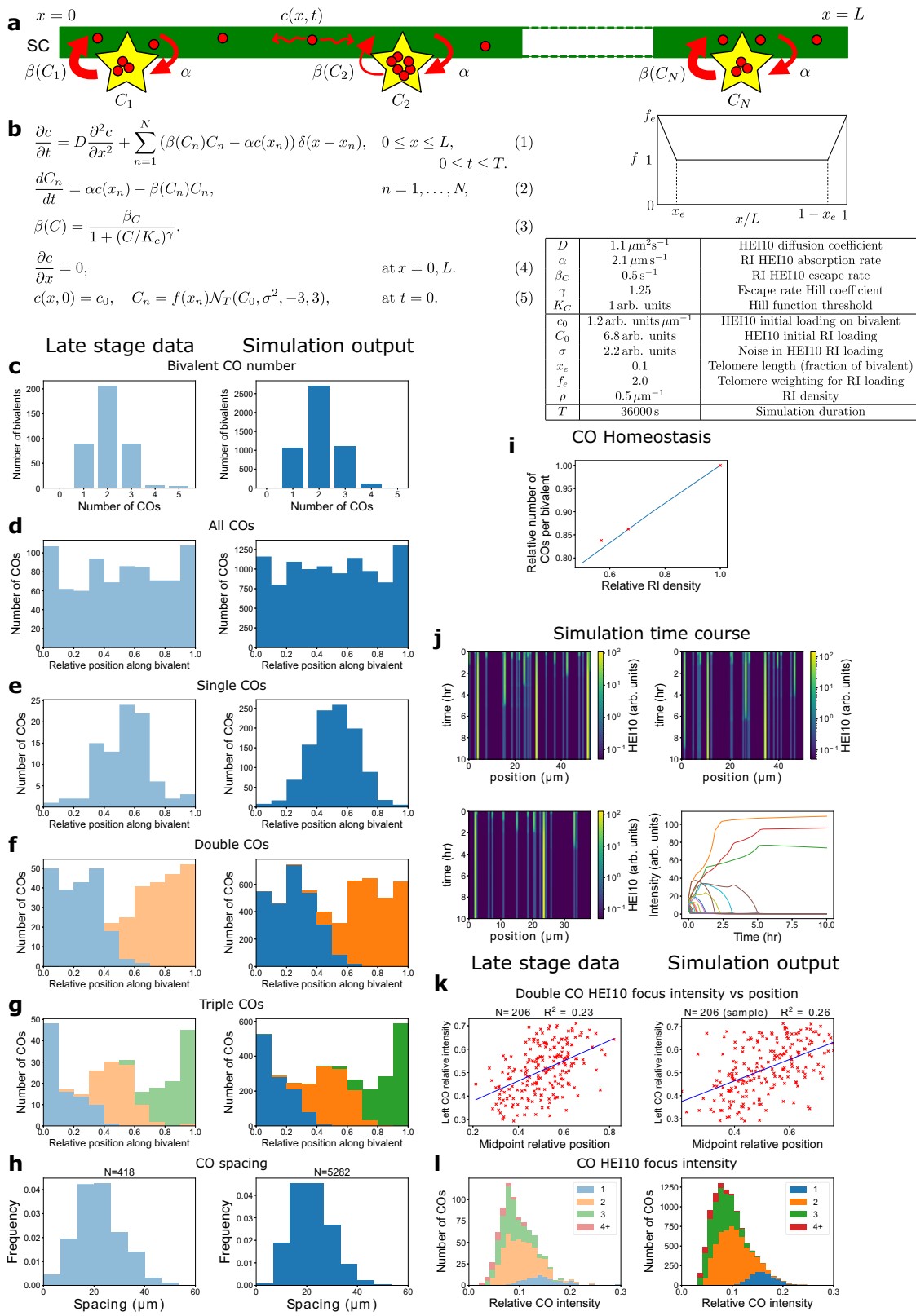

**a**

**b**

$$\frac{\partial c}{\partial t} = D \frac{\partial^2 c}{\partial x^2} + \sum_{n=1}^{N} \left( \beta(C_n)C_n - \alpha c(x_n) \right) \delta(x - x_n), \quad 0 \le x \le L, \tag{1}$$
$$0 \le t \le T.$$

$$\frac{dC_n}{dt} = \alpha c(x_n) - \beta(C_n)C_n, \qquad n = 1, \dots, N, \tag{2}$$

$$\beta(C) = \frac{\beta_C}{1 + (C/K_c)^\gamma}. \tag{3}$$

$$\frac{\partial c}{\partial x} = 0, \qquad \text{at } x = 0, L. \tag{4}$$

$$c(x,0) = c_0, \quad C_n = f(x_n)\mathcal{N}_T(C_0, \sigma^2, -3, 3), \qquad \text{at } t = 0. \tag{5}$$

| | | |
|---|---|---|
| $D$ | $1.1\,\mu m^2 s^{-1}$ | HEI10 diffusion coefficient |
| $\alpha$ | $2.1\,\mu m\,s^{-1}$ | RI HEI10 absorption rate |
| $\beta_C$ | $0.5\,s^{-1}$ | RI HEI10 escape rate |
| $\gamma$ | $1.25$ | Escape rate Hill coefficient |
| $K_C$ | $1$ arb. units | Hill function threshold |
| $c_0$ | $1.2$ arb. units $\mu m^{-1}$ | HEI10 initial loading on bivalent |
| $C_0$ | $6.8$ arb. units | HEI10 initial RI loading |
| $\sigma$ | $2.2$ arb. units | Noise in HEI10 RI loading |
| $x_e$ | $0.1$ | Telomere length (fraction of bivalent) |
| $f_e$ | $2.0$ | Telomere weighting for RI loading |
| $\rho$ | $0.5\,\mu m^{-1}$ | RI density |
| $T$ | $36000\,s$ | Simulation duration |

**Late stage data**   **Simulation output**

**c** Bivalent CO number

**d** All COs

**e** Single COs

**f** Double COs

**g** Triple COs

**h** CO spacing

**i** CO Homeostasis

**j** Simulation time course

**Late stage data**   **Simulation output**

**k** Double CO HEI10 focus intensity vs position

**l** CO HEI10 focus intensity

demonstrated that a ~40% reduction in SC length per cell compared with male meiocytes[19] is accompanied by an equivalent ~40% reduction in CO number per cell, with a shift in COs away from the telomeres[20]. Consistent with these findings, we found that reducing SC length in our model (with otherwise default parameters, including an RI density of 1 RI per 2 µm of SC) by 40% resulted in a 36% reduction in CO number per cell, with a

shift in global CO positioning away from chromosome ends (Supplementary Fig. 4), in full agreement with experiments. Within our own experimental dataset, we also identified a tendency for longer bivalents (measured in µm of SC length) to have more COs, with a comparable relationship identified in model simulations (Supplementary Fig. 5), further supporting a link between SC length and CO number in *A. thaliana*.

**Fig. 2 Comparison between experiments and coarsening model. a**, **b** Model schematic: HEI10 molecules (with 1D concentration $c$) (red) diffuse along SC (green), absorb into foci (yellow, with HE10 amounts $C_n$ at rate $\alpha$, escape foci onto SC at rate $\beta$ (Eqs. (1–3)). Boundary, initial conditions (Eqs. (4–5)). Initial HEI10 amount at RI focus drawn from truncated normal distribution $N_T$ (mean $C_0$, std dev $\sigma$, values restricted within $\pm 3\sigma$)). The graph shows function $f$ modulating initial HEI10 RI loading near bivalent ends. Table: model parameter values. **c**–**h** Comparison of late-stage bivalent cytology data (left) with simulations (right) (experimental data from 395 bivalents from 95 individual cells from 11 wild-type plants). Crossovers (COs) in the experimental data are defined as the positions of late-HEI10 foci. **c** Number of COs per bivalent. **d**–**g** CO positions along bivalents. (813 COs). **d** All COs. **e** Bivalents with single CO. **f** Stacked histograms for double COs, showing positions of 1st/2nd CO. **g** Triple COs, showing positions of 1st/2nd/3rd CO. **h** Distribution of spacing between adjacent COs. **i** Effect of varying initial RI density on CO number per bivalent relative to wild-type (data[18]: red crosses; model prediction: blue line). **j** Example simulation kymographs of HEI10 focus intensities for triple (top left), double (top right), single (bottom left) COs. Shading: $\log_e$ focus intensity. Traces of foci intensity against time (bottom right), showing triple CO simulation (top left). **k** Analysis of double CO HEI10 focus intensities; experiments (left), simulations (right): On arbitrarily aligned SC, intensity of left CO, relative to sum of both intensities, against relative position (measured from left end) of the midpoint between the two COs (CO 'catchment area'). Blue lines: linear-regression best-fits. **l** Histograms of intensities of CO HEI10 foci, relative to sum of all foci within a cell, colour-grouped according to CO number per bivalent (inset); experiments (left), simulations (right). Source data are provided as a source data file.

## HEI10 patterning in lines with variable HEI10 dosage.
After fitting and validating model parameters using wild-type data, we next tested how cytological data obtained from lines exhibiting variable HEI10 dosage compared with model predictions. For these cases, all earlier parameter values were fixed except for those parameters ($c_0$, $C_0$ and $\sigma$, Fig. 2b) specifying the initial concentrations of HEI10, both on the SC and at the RI, which were each multiplied by a constant factor for each variable dosage line. Cytological late-HEI10 focus positioning data were obtained from the well-characterised HEI10 over-expressor line C2[6] and from a *hei10-2 +/−* heterozygous T-DNA insertion line, using the same quantitative imaging pipeline used for the wild-type (Figs. 3a and 4a, Supplementary Fig. 1). We detected an average of 19.9 (s.d. ± 3.7, $n = 33$) and 8.6 (s.d. ± 2.3, $n = 40$) late-HEI10 foci per cell, and 239 μm (s.d. ± 35 μm, $n = 68$) and 237 μm (s.d. ± 32 μm, $n = 62$) total SC length per cell, in the HEI10 over-expressor and *hei10-2 +/−* heterozygous lines, respectively. The best fit for the over-expressor was found with a 4.5-fold increase in HEI10, broadly consistent with measured mRNA levels in HEI10 over-expressing lines[6]. Overexpressor simulations were then a close match for cytological observations of late-HEI10 focus number, positioning and relative intensity (Fig. 3). In particular, the model could capture a much greater number of bivalents experiencing >4 COs, and that COs were positioned much closer together whilst still experiencing weak interference (compare Fig. 3e to Fig. 2h). For the *hei10 +/−* lines, the HEI10 cytological data could again be well fitted to the model across a range of CO measures (Fig. 4) using a 40% reduction in HEI10. For example, the model successfully predicted a much greater number of bivalents experiencing a single CO (but still with no zero-CO bivalents, thus maintaining CO assurance and the obligate CO, Fig. 4b). COs were also found to be generally positioned further apart from one another, thus exhibiting stronger interference, as found cytologically for late-HEI10 foci (compare Fig. 4g to Fig. 3e). Overall, the model fits for cases with variable HEI10 dosage were in good agreement with the data, though with some overrepresentation of HEI10 foci in telomeric regions in our simulations.

## Discussion
In this work, we propose a model whereby diffusion-mediated coarsening dynamics of HEI10 is itself the primary driver of CO positioning and COI in *Arabidopsis*, as speculated over 4 decades ago by Robin Holliday[21]. Our results also demonstrate that COI strength can be tuned simply by altering the cellular abundance of HEI10 in *A. thaliana*. Our model incorporates dynamical clustering of signalling proteins to generate biomolecular foci with a high local protein concentration and, hence, high signalling activity. Similar dynamics have been described in a variety of different cellular processes[22], such as the well-characterised Wnt signal transduction pathway[23]. Furthermore, related models, in which proteins diffuse in one dimension along DNA, and which contain specific regions where the protein has distinct binding/unbinding behaviour, have been successfully used to model other biological phenomena (e.g. bacterial SpoIIIE[24]).

Note that HEI10 coarsening dynamics in the model occur purely in cis, solely along individual pairs of synapsed bivalents, without further exchange with the nucleoplasm. This prevents competition between non-homologous chromosomes for HEI10 and promotes CO assurance. Based on recent evidence, we propose that the SC may act as a potential candidate for promoting cis-coarsening, with *Arabidopsis zyp1* mutants losing CO interference, CO assurance and the wild-type patterning of HEI10 along the SC in zygotene and early pachytene cells, whilst maintaining the presence of late-HEI10 foci in late prophase cells[25,26]. It has also been shown that male/female differences in CO number are abolished in *Arabidopsis* SC mutants, suggesting that the SC is directly involved in mediating heterochiasmy[25].

As COI is a conserved phenomenon, our coarsening paradigm may, therefore, represent a conserved feature of meiotic crossover positioning in other organisms. Consistent with this hypothesis, HEI10 is a member of a broadly conserved family of RING-finger domain-containing proteins with similar dynamics that function within the class I (COI sensitive) CO pathway[8], including Zip3 in *S. cerevisiae*[27], HEI10 and RNF212 in mammals[28,29], Vilya in *Drosophila*[30] and ZHP-3 in *C. elegans*[31]. Whilst we propose HEI10 coarsening to be a key mechanism for CO positioning, we note that perturbations in proteins that affect any of the parameters or initial conditions within our model may also exert measurable effects on interference. We also emphasise that the coarsening mechanism may act synergistically with other elements, such as mechanical stress, to achieve robust regulation of CO positioning[32,33].

## Methods

**Plant materials**. *Arabidopsis thaliana* lines used in this study were wild-type Col-0, the HEI10 overexpressor-line C2[8] and heterozygotes of the *hei10-2* T-DNA insertion line (Salk_014624). *hei10-2* genotyping was performed using the primers hei10-2-F, hei10-2-R and the LBb1.3 T-DNA left border primer. The presence of the HEI10 transgene in the C2 lines was confirmed using the primers HEI10-F and HEI10-R. All primer sequences are listed in Supplementary Table 1. Plants were grown in controlled environment rooms with 16 h of light (125 mMol cool white) at 20 °C and 8 h of darkness at 16 °C.

**Immunocytology**. Immunostained spreads of *Arabidopsis* pachytene cells were prepared for 3D-SIM imaging as follows[34]. To roughly stage the meiocytes, a single anther from a floral bud was removed and squashed in a drop of water on a clean slide under a coverslip and inspected using brightfield microscopy. Early- and mid-pachytene meiocytes were still stuck together within a meiocyte column, whilst late-pachytene meiocytes had begun to break apart from one-another. More precise

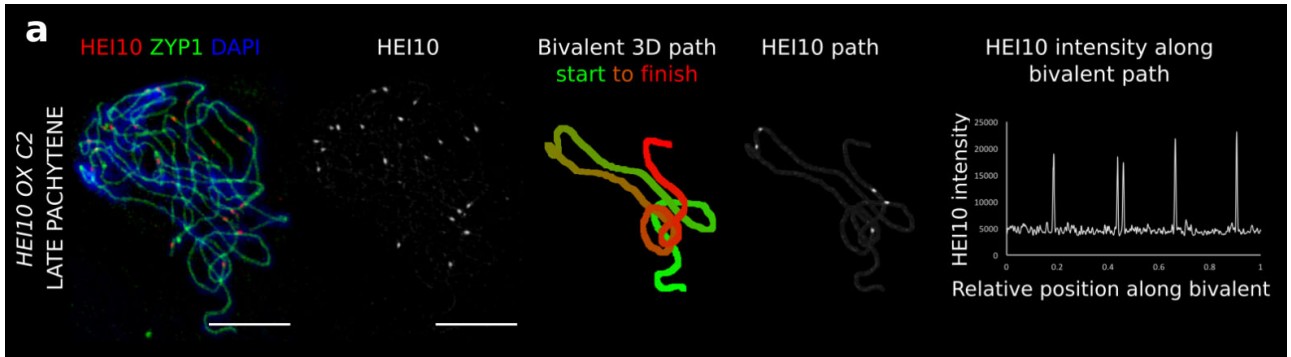

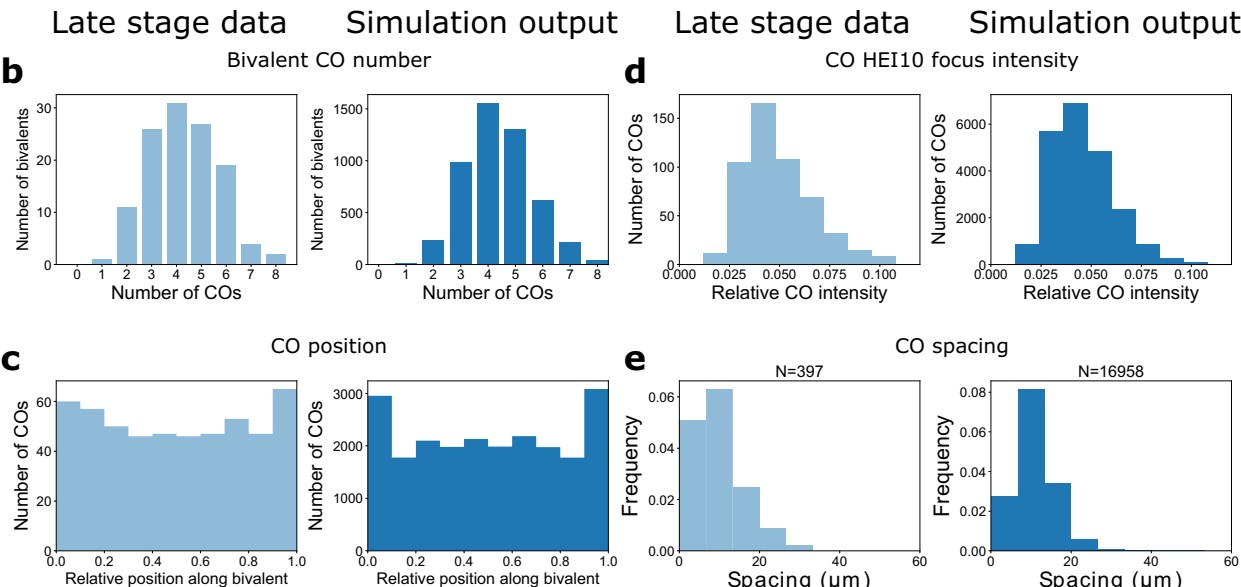

**Fig. 3 Analysis for HEI10 over-expressor. a** 3D-SIM image and example SC measurements from a late pachytene cell of the HEI10 over-expressor (OX) line C2. Representative image shown. Data were collected from 121 late pachytene bivalents for further analysis. Scale bars = 5 μm. **b–e** Experimental data (left) and simulation outputs (right) for HEI10 over-expressor, with simulated HEI10 levels 4.5-times WT (data from 121 bivalents from 32 cells from 4 plants). Crossovers (COs) in the experimental data are defined as the positions of late-HEI10 foci. **b** Number of COs per bivalent. **c** CO positions (all) along bivalents. (Data from 518 COs.) **d** Histograms of intensities of CO HEI10 foci, relative to sum of all foci within a cell. **e** Distribution of spacing between adjacent COs. Source data are provided as a source data file.

staging of early and late pachytene meiocytes was also based on previously defined HEI10 behaviour[8], with mid pachytene meiocytes exhibiting an intermediate phenotype. The remaining 5 anthers containing meiocytes of the desired stage were dissected from the staged buds. They were then macerated using a brass rod on a No. 1.5H coverslip (Marienfeld) in 10 μl digestion medium (0.4% cytohelicase, 1.5% sucrose, 1% polyvinylpyrrolidone in sterile water) for 1 min. Coverslips were then incubated in a moist chamber at 37 °C for 4 min before adding 10 μl of 2% lipsol solution followed by 20 μl 4% paraformaldehyde (pH 8). Coverslips were dried in the fume hood for 3 h, blocked in 0.3% bovine serum albumin in 1× phosphate-buffered saline (PBS) solution and then incubated with primary antibody at 4 °C overnight and secondary antibody at 37 °C for 2 h. In between antibody incubations, coverslips were washed 3 times for 5 min in 1× PBS plus 0.1% Triton X-100. Coverslips were then incubated in 10 μl DAPI (10 μg/ml) for 5 min, washed and mounted on a slide in 7 μl Vectashield. The following primary antibodies were used at 1:500 dilutions: anti-HEI10 (rabbit), anti-ZYP1 (rat) and anti-ASY1 (guinea-pig). The following secondary antibodies were used at 1:200 dilutions: anti-rat Alexa Fluor 488 (RRID AB_2534074), anti-rabbit Alexa Fluor 555 (RRID AB_2535849) and anti-guinea pig Alexa Fluor 647 (RRID AB_2735091). Immunostained cells were imaged using 3D structured illumination microscopy (3D-SIM) on a Zeiss Elyra PS1 microscope equipped with an EM-CCD camera, a Plan-Apochromat ×63, NA 1.40 oil objective and 405, 488, 561 and 642 nm solid-state laser diodes. Slides were imaged in 3D SIM mode with three stripe angles and five phases according to the microscope manufacturer's instructions. Z-stacks were captured at an interval size of 0.0909 μm, with the size of each stack being sufficiently large for the cell to be out of focus at both ends. An immersion oil with a refractive index of 1.515 was used that was optimised for the green/red (ZYP1/

HEI10) channels of our system. To determine the best refractive index, 200 nm TetraSpeck microspheres (Invitrogen) were dried on a coverslip at 1/100 dilution, mounted on a slide in 7 μl Vectashield, and imaged using the same microscope settings as used for the experiments. The symmetry of the point spread function for each channel was then assessed in orthogonal sections through the stack. Channel alignment was performed using the same Tetraspeck beads and the 'affine' alignment algorithm that is included in Zeiss' Zen Black software. Occasional post-processing mismatches in z-alignment were manually adjusted in some cells by trimming the top or bottom stacks to ensure maximal co-localisation of ZYP1 and HEI10 signals to facilitate downstream image analysis. For optimal image quality and to minimise the introduction of reconstruction artefacts, microscope laser power and camera gain values were adjusted for the red (HEI10) and green (ZYP1) channels within a range of 10–30% and 1–3 units, respectively, for each cell to improve contrast and reduce fluorophore bleaching. Bleaching and contrast of raw images was assessed using the SIMcheck plugin to FIJI[35].

**Image analysis**. Each 3D-SIM image contained one nucleus (in a small number of cases multiple nuclei were present, which did not affect the analysis). The image analysis pipeline (Supplementary Fig. 1) contained six main steps: bivalent skeleton tracing, trace fluorescence intensity quantification, HEI10 peak detection, HEI10 foci identification, HEI10 foci intensity quantification, and total bivalent HEI10 intensity quantification. Note that the normalization steps used for foci identification differ from those used for foci intensity quantification; the former was intended to robustly identify foci from noisy traces, whilst the latter was used to carefully quantify foci HEI10 levels.

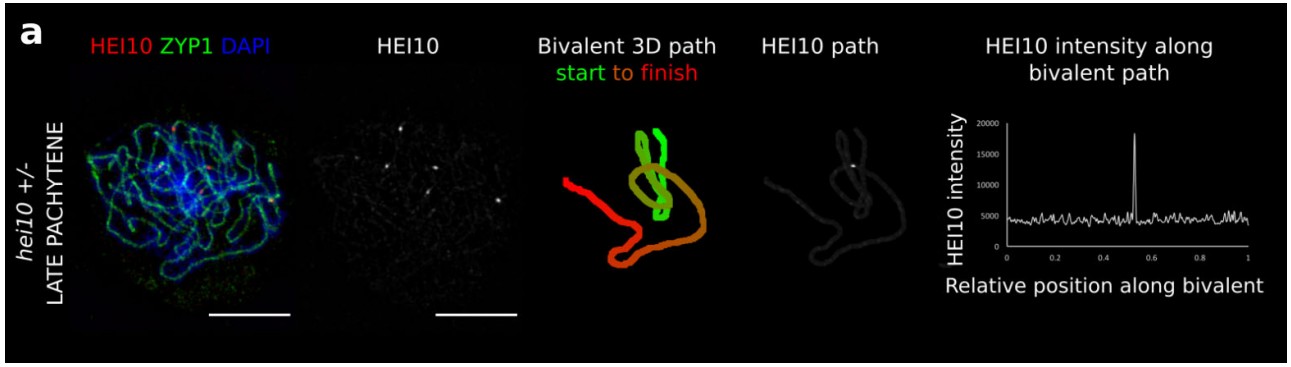

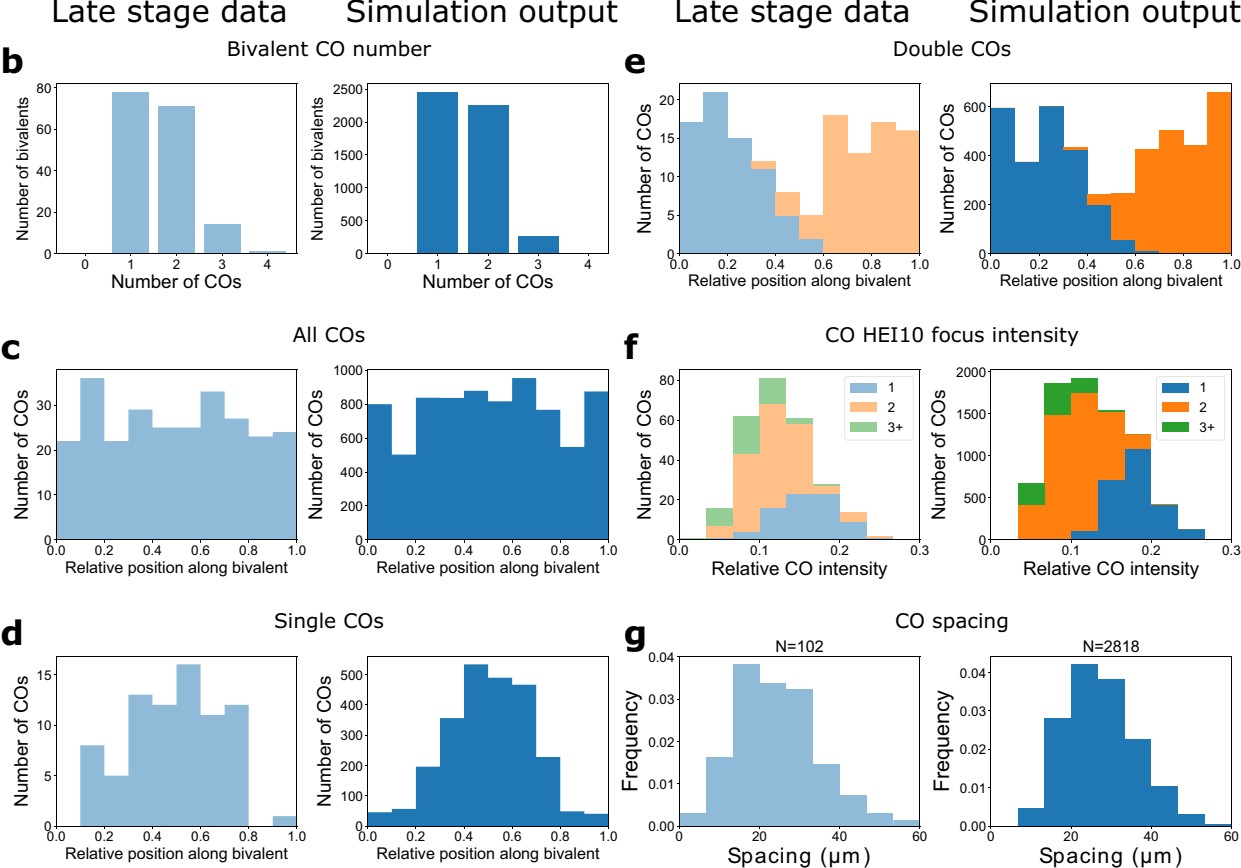

**Fig. 4 Analysis for *hei10* heterozygote. a** 3D-SIM image and example SC measurements from a late pachytene cell of the *hei10-2 +/−* line. Representative image shown. Data were collected from 164 late pachytene bivalents for further analysis. Scale bars = 5 μm. **b–g** Experimental data (left) and simulation outputs (right) for *hei10 +/−*, with simulated HEI10 levels 0.6-times WT (data from 164 bivalents from 40 cells from 3 plants). Crossovers (COs) in the experimental data are defined as the positions of late-HEI10 foci. **b** Number of COs per bivalent. **c–e** CO positions along bivalents. (Data from 266 COs). **c** All COs. **d** Bivalents with single CO. **e** Stacked histograms for double COs, showing positions of 1st/2nd CO. **f** Histograms of intensities of CO HEI10 foci, relative to sum of all foci within a cell, colour-grouped according to CO number per bivalent (inset). **g** Distribution of spacing between adjacent COs. Source data are provided as a source data file.

(1) For bivalent skeleton tracing, individually synapsed pachytene-bivalents were traced and segmented in 3-dimensions from 3D-SIM z-stack images using the Simple Neurite Tracer (SNT) plugin to ImageJ[36]. Traces were made using the linear ZYP1 signal, which localises along the SC. As cells were also stained for ASY1, which localises to asynaptic regions of the meiotic axis, any very small regions of asynapsis that persist into pachytene were incorporated within the measurements to ensure five complete pachytene-bivalents were traced end-to-end in each cell (Supplementary Fig. 1). 3D-skeletons of each pachytene-bivalent path were generated using SNT. Bivalent lengths (in μm) were exported to a Microsoft Excel worksheet for analysis. Further analysis was performed using custom Python scripts [https://github.com/jfozard/HEI10]. Starting from an arbitrary bivalent end, skeleton voxels were ordered. Occasionally, the accidental introduction of branched

segments during the initial SNT tracing step led to the premature truncation of skeleton-traces during the voxel-ordering step and, hence, these traces were excluded from the final analysis.

(2) HEI10 intensities on the path were extracted using the mean intensity from a spherical region of radius 0.2 μm centred at each voxel.

(3) HEI10 intensity trace data (at each point using the average over the spherical region of radius 0.2 μm) was normalized by subtraction of the mean and division by the standard deviation of the HEI10 intensities along the trace. (Note that subtraction of the median, as used in step 4 below, leads to almost identical results). Peaks in the above HEI10 trace intensities were identified using the scipy[37] find_peaks criterion. This criterion identifies local maxima, rejecting peaks whose prominence was less than half of the standard deviation of the HEI10 intensity

values along the trace, or within 5 voxels (along the traced skeleton) of a more intense maximum. Occasionally, small flecks of HEI10 foci from separate, nearby bivalents (intruding foci) were identified within the 0.2 μm spherical regions. These intruding foci usually cause a small peak on the bivalent trace that they intrude upon, but a larger peak on the correct bivalent trace. To eliminate the impact of intruding foci during peak identification we compared absolute trace peak intensities with those on all bivalents within a spheroidal region (10 voxels in each dimension, with the image resolution being 0.04 μm × 0.04 μm × 0.9 μm), rejecting peaks where a larger peak was present within this region.

(4) For foci identification, the same intensity normalization was used as in step 3. Peaks from step 3 were then identified as foci if their normalized intensity was more than 0.4 times the intensity of the largest peak.

(5) For late-HEI10 focus intensity quantification, the median HEI10 intensity along each trace was used as an estimate of background fluorescence and subtracted from the intensities at each focus peak (from step 4). Due to differences in factors such as antibody penetration and local slide conditions, we found that absolute fluorescent intensity values for the same primary and secondary antibody combinations varied considerably between cells, including between cells on the same slide. Therefore, intensities were normalized relative to the sum of all foci intensities within the same cell, in order to minimize the effects of differences in imaging conditions between cells (e.g. Fig. 1b (middle left, middle right)).

(6) To quantify the total HEI10 intensity per bivalent (e.g. for cells earlier in pachytene, where HEI10 foci are difficult to identify—see Fig. 1a upper), the median HEI10 intensity (same value as in step 5) was subtracted from each trace, and then the resulting value integrated along each bivalent trace. Similar to focus intensity quantification (step 5), these intensities were then normalized through division by the sum of the integrated intensities from all bivalents in the same cell.

Further processing of peak intensity and position data were performed using the Python packages imageio, numpy[38], pandas, statsmodels and matplotlib. For the analysis of per bivalent relative total HEI10 intensities in Fig. 1b (left), wild type cells of the appropriate stage ('early', 'mid' or 'late') were included in the analysis. Cells that could not be clearly staged were excluded from the analysis. Relative SC lengths were calculated by dividing each SC length by the total length of the SCs in the cell. SC relative HEI10 intensities were calculated as in step 6 above. For the central two panels of Fig. 1b, analysis was restricted to all 'late' cells for which all bivalents had been successfully traced. Relative foci HEI10 intensities were calculated as in step 5 above and summed over all foci on each SC. Linear regression in Fig. 1b was performed using the Python 'statsmodels' package. Kruskal–Wallis and Dunn's tests to compare the distributions of relative intensities of foci for SCs with different numbers of foci were performed using the R commands 'kruskal.test' and 'dunn.test'. For all data shown in Figs. 2–4, all SCs which were staged as 'late' and which were categorized as being good quality traces were included in the analysis. For the histograms of foci positions, these positions were rescaled to the unit interval by division by the length of the SC, and histograms generated with 10 equally sized bins. For the plots of double CO HEI10 focus intensity against position in Fig. 2k, relative intensities (from step 5 above) were normalized by dividing by the sum of the relative intensities of the two foci on the SC. For the plots of CO HEI10 foci intensities (Figs. 2l, 4f), focus intensities were normalized against the sum of all focus intensities in the same cell, as in step 5.

In order to increase our confidence that the negative relationship identified between late-HEI10 relative focus intensity and late-HEI10 focus number per bivalent did not result from any unexpected bias within the imaging approach described above, we also re-analysed a subset of 10 high-quality wild-type cells using an alternative approach (Supplementary Fig. 2). Late-HEI10 foci were identified from 3D-SIM z-stacks using the FociPicker3D FIJI plugin[39], which allows 3-dimensional foci to be automatically identified against a uniform background intensity threshold. Uniform background intensity thresholds and minimum pixel size values were manually optimised within a range of 15,000–30,000 and 5–10, respectively, for each image to ensure that all late-HEI10 foci were correctly identified, with all otherwise default parameters (Supplementary Fig. 2b). Maximum intensity values were then obtained for each focus and normalized by dividing these values by the summed maximum intensity values of all late-HEI10 foci values within the same cell to give a relative maximum intensity value for each focus. The 5 individual bivalents within a cell were segmented and identified using the 3D-SIM ZYP1 channel and the simple-neurite tracer plugin to FIJI[36] (Supplementary Fig. 2c). Once again, using this alternative method, we identified significant differences in late-HEI10 relative focus intensity for foci on bivalents with different total numbers of late-HEI10 foci (Supplementary Fig. 2f). Additionally, whilst the 3D-SIM image reconstruction method used on the ELYRA microscope is derived from linear algorithms[40] and, hence, should retain relative intensity differences that are present within the original, unreconstructed widefield image, we sought to confirm this by using the same FociPicker3D approach to identify and measure late-HEI10 focal intensity in pseudo-widefield images, generated using the SIMcheck FIJI plugin on the same raw data that was used to generate the 3D-SIM images, of the same subset of 10 wild-type cells (Supplementary Fig. 2a). Once again, uniform background intensity thresholds and minimum pixel size values were manually optimised within a range of 7550–18000 and 5–10, respectively. This confirmed that the relative maximum intensities of late-HEI10 foci from the pseudo-widefield images were linearly correlated with the relative maximum late-HEI10 foci intensities of the same foci in the 3D-SIM images (R = 0.96) (Supplementary Fig. 2d) and, hence, that any relative intensity

differences present in the original widefield images were retained within the 3D-SIM reconstructed images. The same significant differences between relative maximum HEI10 foci intensity and late-HEI10 foci number per bivalent were also detected within the pseudo-widefield data (Supplementary Fig. 2e). A Kruskal–Wallace test was used to identify significant differences in relative maximum HEI10 foci intensity on bivalents with different late-HEI10 foci number (chi-squared = 27.061, df = 2, p-value = 1.33e−06 and chi-squared = 25.465, df = 2, p-value = 2.953e−06 for the pseudo-widefield and 3D-SIM data, respectively, n = 101 foci) followed by a post-hoc Dunn's test of multiple comparisons using rank sums with a Bonferonni correction (Supplementary Fig. 2e, f). The R packages 'kruskal.test' and 'dunn.test' were used.

**Mathematical modelling.** We employed a hybrid stochastic/deterministic simulation approach, where the initial conditions incorporate stochasticity both in the initial loading of HEI10 onto the SCs, and in the length of the SC, but thereafter the dynamics follow a deterministic trajectory specified by differential equations. The initial stochasticity is essential given the observed highly variable nature of the HEI10 loading in our experiments. As these initial conditions are stochastic, multiple realisations were simulated (1000 for each chromosome length). For the subset of the late-stage WT data for which all the bivalents within each cell were well-traced, we sorted the bivalent skeleton traces (described in 'Image analysis' section) by length, and used these lengths to estimate the empirical distributions of each of the first to fifth longest bivalents within each cell. Note that, in the absence of another marker, we were unable to identify SCs with specific chromosomes other than through their lengths. Mean and standard deviation parameters of these distributions are listed in Supplementary Table 2. First, for each of the five SC lengths (first to fifth longest SC within a cell), a length L was generated at random from the appropriate empirical distribution, approximated by a normal distribution (with values clamped to the range within three standard deviations of the mean). For each sampled L, the number N of RIs was chosen to give 1 RI per 2 μm, rounded to the nearest integer, consistent with our SC measurements and experimentally measured RI numbers[12], and early HEI10 foci numbers[8] per cell. These N RIs were then placed at random positions, sampled from a uniform distribution on the length of the SC, and sorted in order of their position along the SC. The RI positions were denoted by $x_1 < x_2 < \cdots < x_N$, along the SC, where $x$ indicates distance along the SC. At the start of the simulation, a random amount of HEI10 was loaded at each focus, drawn from a Gaussian distribution (truncated at −3 and +3 standard deviations, to ensure nonnegativity of initial conditions). In addition, a uniform concentration of HEI10 was loaded on the SC in between foci. To account for earlier synapsis at telomeres[16], and to recapitulate the increased late HEI10-foci density at SC ends in the WT data, we multiplied the initial foci HEI10 loading by a non-uniform, piecewise linear function $f(x/L)$ as shown in Fig. 2b. This function took its highest value $f_e = 2$ at SC ends, and decreased linearly with distance to a constant value of 1, at a distance 0.1 L from the SC ends. We ran 1000 realisations for each of the 5 empirical SC length distributions. To test for robustness, we also simulated a version of the model with an exponentially decaying form for $f(x/L)$ with position (giving similar results) and also without the non-uniform function $f(x/L)$ (Supplementary Fig. 3). In the latter case, we found similar overall results, although the simplified model could not adequately recapitulate a high enough density of COs close to the chromosome ends for the 2 and 3 CO cases.

During the dynamical simulations, HEI10 underwent one-dimensional diffusion along the SC axis, with a diffusion coefficient D. At the RIs, HEI10 could be absorbed into foci, at a constant rate α, and escaped back onto the SC at a rate β This escape rate depended on the amount of HEI10 at each focus. These considerations specified the system shown in Eqs. (1) and (2) of Fig. 2b, for the one-dimensional concentration $c(x,t)$ of SC HEI10, and the HEI10 amounts at the foci $C_n(t)$, $n = 1, ..., N$. The flux of HEI10 from the SC to the foci, and the reverse, is accounted for by point sink/source terms in Eq. (1). The functional form for β is given in Eq. (3), where the parameter $\gamma > 1$ controls how rapidly the escape rate diminishes for increasing $C_n$. No flux boundary conditions (4) were imposed at SC ends. Note that, by construction, this system of equations conserves the total amount of HEI10 on each SC.

Eq. (1, see Fig. 2b) was discretized by dividing each SC into M = 2000 compartments of length $h = L/M$, with a compartment HEI10 concentration given by $c_m(t)$ where $m = 1, \ldots, M$ and replacing the spatial second derivative by its standard finite-volume approximation (see e.g. ref. [41]), giving

$$\frac{dc_1}{dt} = D\frac{c_2 - c_1}{h^2} + \frac{1}{h}\left(\sum_{n\,s.t.\,0 \leq x_n < h}(\beta(C_n)C_n - \alpha c_1)\right) \quad (6)$$

$$\frac{dc_m}{dt} = D\frac{c_{m-1} - 2c_m + c_{m+1}}{h^2} + \frac{1}{h}\left(\sum_{n\,s.t.\,(m-1)h \leq x_n < mh}(\beta(C_n)C_n - \alpha c_m)\right) \quad (7)$$

$$\frac{dc_M}{dt} = D\frac{c_{M-1} - c_M}{h^2} + \frac{1}{h}\left(\sum_{n\,s.t.\,(M-1)h \leq x_n \leq L}(\beta(C_n)C_n - \alpha c_M)\right) \quad (8)$$

Combined with Eq. (2, see Fig. 2b), these equations comprise a system of ordinary differential equations, for $c_m$ and $C_n$. This system was solved using the Rodas5[42] Rosenbrock method, as implemented in the DifferentialEquations.jl[43]

Julia package. The equations were integrated over a period $0 \leq t \leq T$, with the parameter $T = 36,000\,\mathrm{s} = 10\,\mathrm{h}$. This numerical integration method used an adaptive timestep with relative error tolerance of $10^{-12}$, and a non-negativity constraint on all variables at intermediate integration timepoints. Foci were identified as COs using the same criterion as for the experimental data (i.e. if their intensity was more than 0.4 times the intensity of the brightest focus).

The diffusion coefficient, $D = 1.1\,\mu\mathrm{m}^2\,\mathrm{s}^{-1}$, was chosen to be compatible with measurements of the diffusion coefficient for an analogous protein, ZHP-3, in *C. elegans*[13]. The remaining default parameters are listed in Fig. 2b. These values were fitted through comparison of simulation outcomes with our experimental data. Because of the model simplicity, we were able to find these parameter values through manual exploration of simulation outputs without the need for automated parameter searches. To assess robustness to parameter value choices, simulations were performed in which each parameter value was perturbed by 10%, giving broadly comparable results.

We also developed a version of the model for female bivalents (Supplementary Fig. 4). For all female bivalent length distributions, the mean and the standard deviation were reduced by 40% compared to their male counterparts.

We also developed a version of the model with an alternative form for the RI escape rate, namely

$$\beta(C) = \beta_C / (1 + \delta e^{C/K_C}) \tag{9}$$

With appropriately modified parameter values, we were again able to obtain a reasonable fit to the wild-type data (Supplementary Fig. 6). We note that an important feature of this escape rate is that it decreases faster than $1/C$ for large $C$, so that RIs with high levels of HEI10 outcompete those with lower levels. However, if the escape rate decreases too quickly, coarsening stalls and too many foci are generated at the end of pachytene.

The model captures the phenomenon of the 'obligatory crossover', with very few SCs having zero COs at the end of the simulation. Such robust behaviour is a natural property of coarsening models and motivated our initial model development. Provided that the initial HEI10 loading is sufficiently high, HEI10 foci exceeding a critical threshold will form and grow at the expense of the smaller foci, concentrating HEI10 into progressively smaller numbers of foci. However, the focus number cannot drop below one, as after coarsening into a single focus, that focus is necessarily stable, as there is then nowhere else for HEI10 to accumulate. At sufficiently low HEI10 levels the model will fail to coarsen, and instead will be attracted to the (now stable) steady state with equal levels of HEI10 at each RI.

In the model, HEI10 can bind back and forth between the foci and a freely diffusible form on the SC. We emphasise that this change in location will almost certainly involve accessory proteins and modification of the HEI10 protein itself to make such transitions energetically possible. However, such accessory proteins could simply be RI and/or SC localised and therefore not show any further dynamical patterning. We have, therefore, not explicitly included such proteins in the simulations. They are instead treated implicitly as allowing successive cycles of HEI10 localisation and release from HEI10 foci. It is possible that these accessory proteins could play a more dynamic patterning role, for example, by functioning as a repressor of HEI10 away from the foci. They could then exhibit coarsening behaviour along with HEI10. Our model could easily be generalised into such a form. However, such proteins are clearly not logically required to explain CO positioning, as our model, which lacks such explicit repressive factors, can explain essentially all our current data. Furthermore, there is currently little evidence for the presence of such proteins, at least in *A. thaliana*.

**Reporting summary**. Further information on research design is available in the Nature Research Reporting Summary linked to this article.

## Data availability
Data supporting the findings of this work are available within the paper and its Supplementary Information files. Imaging data used in this study, which is associated with all figures, was deposited to the Image Data Resource (https://idr.openmicroscopy.org) under accession number idr0107. Source data are provided with this paper.

## Code availability
Custom Python scripts for data analysis, and Julia code to perform numerical simulation of the model, is available at https://github.com/jfozard/HEI10 (https://doi.org/10.5281/zenodo.5076176).

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

## Acknowledgements

We gratefully acknowledge E. Coen, F.C.H. Franklin and members of the X. Feng lab for fruitful discussions on this topic, F.C.H. Franklin for supplying the primary antibodies and E. Wegel for microscopy support. This work was supported by BBSRC Institute Strategic Programme GEN (BB/P013511/1) to M. Howard and K.B, European Research Council Consolidator grant (CoG EVO-MEIO 681946) to K.B., and JIC Institute Strategy Fund grant CX516F13A to C.M.

## Author contributions

C.M., J.A.F. and M. H. jointly conceived the study, and designed the mathematical model. C.M. performed the experiments, imaging, and the initial data analysis. J.A.F, C.M. and M. H. developed the image processing pipeline. J.A.F. performed numerical simulations and analysis. I.R.H. provided plant materials. C.M., J.A.F., M. H., M. H. and K.B. wrote the paper.

## Competing interests

The authors declare no competing interests.
