## [Peer Review File · Nature Communications]

REVIEWER COMMENTS

Reviewer #1 (Remarks to the Author):

This manuscript is dealing with the long-standing question of the regulation of crossover (CO) distribution along chromosomes. It is known for a long time that CO numbers and positions are very tightly controlled but the mechanism(s) involved is (are) still very obscure. Here the authors propose that CO position is determined by the diffusion along a given bivalent/SC complement of a certain amount of the HEI10 protein that ends up accumulated to a limited number of sites that will become COs.

This manuscript relies on an impressive amount of image analyses and the authors present a very interesting working hypothesis on how Class I COs could become designated. I like the idea of a chromosome/bivalent/SC complement intrinsic control. However, I cannot say the data presented here support it for the several reasons: (i) Experimental procedures and image analysis workflows are very difficult to understand, putting the reader in the position to guess what the authors have done. This is of course unacceptable and jeopardize considerably the credibility of the approaches. (ii) Data are hardly analyzed. Instead the manuscript jumps directly to modellization and leaves behind any thorough analysis of the collected data. (iii) I have serious concerns about the chosen experimental procedures for HEI10 intensity measurements (see below)

Major points:

- The image analysis procedure: Fluorescent image quantification is a difficult point and requires very rigorous experimental design to be achieved properly. Here, I cannot see that the authors paid attention to that but instead they chose to work after the acquisitions on the normalization of the fluorescent signals. This is even more of a problem here, since the authors use SIM microscopy which is based on structured illumination and image reconstruction. This technique is particularly prone to artifacts and introduces nonlinear modifications of the signals that, to my opinion, are hardly compatible with intensity quantification. If I have no doubt that their procedure allows the identification and mapping of HEI10 foci on SC, I'm not sure (to say the least) that intensity quantification is relevant. I think quantification of SIM images requires certain calibration of the experiments which do not appear anywhere here.

- Issues with the normalization procedures:

1. They are particularly difficult to understand.
2. It seems that there have been several layers of normalization of the HEI10 signal (each involving subtraction of "background", cut off etc... I'm not sure that, at the end of such treatment (several normalization rounds based on different parameters), that statistical analyses can be done.
3. From my point of view, it is highly questionable to normalize by HEI10 intensity. Above all considering that the whole point of the study is centered around modulation of HEI10 quantity along chromosomes...If the authors want to understand how HEI10 quantity is evolving, they should have normalized by another cellular component. DAPI intensity? ZYP1 signal? Axial element labeling? I cannot understand why these analyses haven't been done. They would allow to compare HEI10 intensities between two cells even at different pachytene stages (early/middle/late) and validate or not that HEI10 intensity is constant from early to late pachytene, which if I understand properly is one of the founding hypotheses of their model.

According to the Materials and Methods section, authors can stage cells from one anther and use the remaining 5 to target a specific pachytene stage. So, I suggest that they mix stages on slides to prevent slide variability and be able to compare HEI10 intensity on cells at different pachytene stages. Acquisition on the same slide and with same exposure time will allow them to compare HEI10 intensity either with SIM or with epifluorescence. For me this is the only way to prove what is written line 130: "Overall, the initial amount of HEI10 in the model is consistent with a fixed amount per μm of SC, as observed cytologically (Fig. 1b)." How things have been done here, there is absolutely no way to

compare early and late HEI10 intensity so this assumption is definitely not true.

4. Nowhere in the text is it explained how are treated the data once normalized. The reader gets the impression that all data extracted from different cells are pooled together. If it is the case it makes absolutely no sense to compare intensity. Therefore, I guess (literally, because I cannot find the information anywhere) that the HEI10 focus intensities given are corresponding to a ratio, related to total cell intensity. This has to be explained properly.

-Data analyses: I don't understand why the collected data are so poorly analyzed. Hardly any quantitative analyze of the data is provided before the modelling part.

What is the average number of HEI10 late foci per cell? Do these numbers fit with already known data?

The authors say that HEI10 foci intensity is stronger when HEI10 foci are single than when multiple. What is the range of intensity variation we are talking of? Is it true whatever is the total amount of HEI10 foci per cell?

- Another important point is to understand how the authors cope with the timing of appearance of HEI10 late foci? Could the different of intensity they detect reflect a difference in timing? The authors treat their data of late HEI10 foci as if all the cells had reached a final state but instead it is very likely that their late pachytene cells contain an heterogenous population of cells. How far the differences of intensity they detect could reflect a different in the stage at which the cells are?

- The negative correlation between HEI10 intensity and HEI10 foci number is a very strong statement in this work. I think the authors absolutely need to strengthen it.

1) I suggest that they repeat HEI10 intensity measurement using epifluorescence microscopy imaging and proper image acquisition calibrations.

2) If the authors are right about the negative correlation between HEI10 focus intensity and number of foci per bivalent and that this difference is not explained by a staging issue, they should find the same correlation at later stages (diakinesis) when they are sure to look at mature HEI10 foci. This is a control relatively easy to achieve.

- Line 162/166: it is stated that concerning the case of double CO "the size of the leftmost of the late-HEI10 intensity peak was generally greater when the rightmost peak was positioned close to the right bivalent end" and that this result of the model was confirmed on the cytological data".... But I really cannot see how left and right end of a SC can be defined on bivalents that have no orientations!

Minor points:

- I129: "initial amounts of HEI10 at foci near bivalent ends were greater than for those elsewhere (Fig. 2b)." I don't see anything like that on 2b.

- I127: "To account for the preferential initiation of synapsis near telomeres"
Telomere driven synapsis evidence are very scarce in *A. thaliana* as far as I know

- The authors should be careful not to use "CO" for "HEI10 foci". It's a shortcut that can very well turn wrong, above all when analyzing pachytene HEI10 foci

Reviewer #2 (Remarks to the Author):

As Morgan et al. notice, CO patterning is poorly understood, particularly as it comes to elucidating CO interference. The authors propose a neat mathematical model that relies on cytological analysis of HEI10, a key protein involved in the progression of recombination intermediates to COs. Using this

model, the authors can predict CO distribution in wild-type Arabidopsis as well as mutant lines with reduced and increased expression of HEI10. Overall, this is an interesting piece of work. There are a few areas where the manuscript needs improvement.

1. I worry that the underlying mechanistic explanation on HEI10 protein diffusion along chromosomes is very speculative. This process has not been experimentally observed and there may be many questions on how it could actually work in the context of chromosomes and chromatin. Is there any experimental evidence the authors could cite in support of this model?

2. Why would sites with more HEI10 protein be more likely to generate COs?

3. The authors could consider examining HEI10 localization in a line with drastically reduced DSB numbers. If the presence of large vs small foci is dependent on the amount of HEI10 protein being produced, only large foci should be found.

4. Is there evidence for CO numbers correlating with SC length in Arabidopsis other than the male-female relationship? As male-female recombination differences are not that well understood at the molecular level, it might be that SC length is not the only or not the most important difference. Can the authors examine the lines from Xue et al. to see whether they exhibit shorter SC?

5. Line 67-68: "widely varying HEI10 intensity as a function of position in μm along the SC" –is there a possibility that some of this intensity variation is caused by variable accessibility of HEI10 molecules to the antibody rather than just the HEI10 protein amount? Is there a way to normalize HEI10 fluorescence, for example by immuno-colocalizing with ASY1, H1 histone or anything else that should show uniform staining along chromosomes?

6. Line 128: Does synapsis in Arabidopsis indeed start exclusively at telomeres? I thought there were also data indicating that synapsis initiation was interstitial.

7. Lines 175-178: Could the authors elaborate on how exactly this simulation was done? Did it assume the same number of DSB/RI in male and female meiosis? Is this a fair assumption?

8. Lines 149-152: Could the authors elaborate on the added benefit of their analysis in this case. Xue et al. already explained the extent of CO reduction by just considering the need for an obligate CO and the correlation between COs and DSBs.

9. Lines 183-185 and on: Rather than speculating, the authors could determine this pattern using biological data, at least for the pericentromeric regions. This can be done, for example, by colocalizing centromere proteins with either Hei10 or DMC1/RAD51.

Minor issues:

1. Line 25: I would suggest "crossover positions can be identified" rather than "explained"

2. Line 85: "SC length dependent manner": this phrase is not clear. Could the authors elaborate?

Reviewer #3 (Remarks to the Author):

The manuscript entitled "Diffusion-mediated HEI10 coarsening 1 regulates meiotic crossover positioning" by Chris Morgan and colleagues addresses the mechanism responsible for the crossover between sister chromosomes observed during meiosis and in particular why all chromosomes must receive at least one "obligatory crossover" while crossovers are prevented from occurring near one

another by "crossover interference". Although this phenomenon is known for more than 100 years, the reasons behind it are still unclear and in this study, the authors investigate this problem by using quantitative super-resolution cytogenetics and mathematical modelling in *Arabidopsis thaliana*.

The authors showed that crossover positions could be explained by a predictive, diffusion-mediated coarsening model, in which large, approximately evenly-spaced HEI10 foci grow at the expense of smaller, closely-spaced clusters. They interpret crossover interference observed in *Arabidopsis* in terms of the coarsening process of the model. Their model recapitulated the "gain of function" and "loss of function" phenotypes observed in the *Arabidopsis thaliana* wild-type with the over-expressor of the conserved E3 ligase HEI10 and a *hei10* heterozygous line5 model, respectively, by changing one single model parameter, the HEI10 dosage.

Crossover between homologous chromosomes is of fundamental importance to create genetic diversity within offspring. This study puts forward a plausible mechanism implemented in simple and yet elegant mathematical model which is quantitatively in agreement with the presented super-resolution cytogenetics data. The manuscript is concisely and nicely written. That being said, I found the following points important to address.

Majors:

1. The authors investigated this problem in *Arabidopsis thaliana*, although as their findings might be extrapolated to other eukaryotes (as they indeed speculated), the manuscript title should reflect the name of this organism.
2. The authors did not demonstrate that diffusion is the mechanism controlling HEI10 dynamics, which in turn would regulate meiotic crossover positioning. Instead, they conceived a model assuming diffusion of HEI10, which nicely recapitulates their experimental results. Hence, the agreement between the model and the experimental data suggests that spreading of HEI10 could be in fact driven by diffusion. Nevertheless, other not yet tested mechanisms could also be at work. Thus, as tempting as it might be to invoke this mechanism, I believe that some caution could be exercised in the article's title.
3. The authors find that HEI10 levels correlate with SC lengths. Nevertheless, this correlation is not obvious by visual inspection of the plots (Fig. 1 b) and a measure of this linear correlation is missing.
4. Regarding the initial condition for the SC lengths, some effort could be done to better explain how these lengths are determined.
 - a) In the Extended Data Fig. 5, the authors showed a table of SC lengths used for simulations. It is not clear to me what is the origin of these data. Are these values measured by the authors? If so, they should stay so, if not, a proper reference should be added.
 - b) Between lines 101-104 of the main text, the authors wrote "a synapsed pachytene bivalent is considered as a one-dimensional structure, with the simulated SC length drawn at random from the experimental bivalent SC length distributions". Then, in the Methods section "Mathematical modelling" below line 361, the authors stay "where the initial conditions incorporate stochasticity [...] in the length of the SC" and below, in the same section, "for each of the five SC lengths, a length L was generated at random from the appropriate empirical distribution, approximated by a normal distribution (with values clamped to the range within three standard deviations of the mean)". If I understood correctly, the authors modeled the 5 SC lengths (from the 5 chromosomes) with 5 normal distributions, each of them parametrized with the mu and sigma extracted from the table?
5. How exactly the initial conditions for RI were implemented? The authors wrote (main text, lines 104-106) "RIs were placed at random positions along the five bivalents, in accordance with the cytologically defined random or weakly-interfering nature of early recombination nodules in plants". In lines 373-377, methods section, they wrote "RIs were then placed at random positions x_1, x_2, \dots, x_N ,

along the SC, where x denotes distance along the SC. For each SC with randomly-generated length, the number of RIs was chosen to give 1 RI per 2 μm , rounded to the nearest integer". What is the statistical distribution of the RI?

6. The authors consider that HEI10 escapes from the immobile foci back onto the SC at a rate β , which they assume is the popular Hill-expression. Regardless the fact that maybe more effort could be made to make clear why this expression was chosen, an important question is how robust is the model to this assumption. What would happened if other expressions were tested?

7. The authors wrote (Lines 193-196) "After fitting and validating model parameters using wild-type data, we next tested how cytological data obtained from lines exhibiting variable HEI10 dosage compared with model predictions. For these cases, all earlier parameter values were fixed except the HEI10 concentration, which was treated as a single fitting parameter". First, I guess the authors are referring to the initial concentration of HEI10 (c_0 in the model), so it would be better if the authors would explicitly stay that. Second, and more important, by considering c_0 as the only fitting parameters, the authors are assuming that the experimental conditions behind HEI10 over-expressor line C2 and the *hei10-2* +/- heterozygous T-DNA insertion line would be equivalent to the wild-type with the only exception of this parameter. Although this assumption is certainly straightforward and reasonable, I am wondering whether, for instance, the parameters reflecting the initial RI loading in these two experimental conditions could be different from the wild type condition as well. Why the authors discarded this possibility and others?

8. (Lines 208-209) The fact that the bivalent CO number was zero (Fig. 2 c right, Fig. 3 b right, Fig. 4 b right) is an emergent result of the simulations or a constrain imposed by the authors. If it is the first one, it is remarkable and should be made explicit and if it is the second, it should be made explicit.

9. The mathematical model here proposed by the authors has some resemblance with one we reported two years ago. In that model, a molecular motor (SpoIIIE) is able to bound/unbound to a DNA molecule, undergoes 1D-diffusion along the DNA molecule and changes their diffusion coefficient and binding/unbinding rates in a "special region" of the DNA molecule (SRS sequence) (Chara et al., 2018. Sequence-dependent catalytic regulation of the SpoIIIE motor activity ensures directionality of DNA translocation. *Sci Rep.* 8, 5254. <https://doi.org/10.1038/s41598-018-23400-8>). Although the here proposed model involves more than one "special regions" (RIs in this model) and, all in all, is addressing a different biological phenomenon, acknowledging the models resemblance could strength the current model plausibility since this modelling strategy proved to be successful in other scenario.

Minors:

In Fig. 1 b and its legend, the concentration of HEI10 along the SC (c) is not explicitly defined.

Legend Fig. 2 i. In the figure, we see red crosses and in the legend are referred as red circles.

The authors stay "We emphasize that the model is simple, with only five dynamical parameters and another five specifying initial conditions (parameter values in Fig. 2b and Methods)". First, I might be wrong but I count in Fig. 2b 6 parameters corresponding to the initial conditions, and second, although I understand the point, I think it is at least debatable to characterize an 11-parameters model as "simple".

Prof. Dr. Osvaldo Chara

Dear Chuanfu,

Thank you for forwarding the reviewer comments. We would also like to thank the reviewers for their great diligence in commenting on our work. The reviewers raised many constructive points that have led us to perform additional experiments and analysis, and have helped us improve the manuscript presentation. We believe that we have been able to address all of these points and we therefore hope that the manuscript will now be found suitable for publication in Nature Communications. We also note that you have requested bar charts to be replaced within the figures. However, we have retained the bar charts (histograms) as we believe they accurately define the distribution of the underlying data.

Kind regards,
Martin Howard.

Reviewer #1 (Remarks to the Author):

This manuscript is dealing with the long-standing question of the regulation of crossover (CO) distribution along chromosomes. It is known for a long time that CO numbers and positions are very tightly controlled but the mechanism(s) involved is (are) still very obscure. Here the authors propose that CO position is determined by the diffusion along a given bivalent/SC complement of a certain amount of the HEI10 protein that ends up accumulated to a limited number of sites that will become COs.

This manuscript relies on an impressive amount of image analyses and the authors present a very interesting working hypothesis on how Class I COs could become designated. I like the idea of a chromosome/bivalent/SC complement intrinsic control. However, I cannot say the data presented here support it for the several reasons: (i) Experimental procedures and image analysis workflows are very difficult to understand, putting the reader in the position to guess what the authors have done. This is of course unacceptable and jeopardize considerably the credibility of the approaches. (ii) Data are hardly analyzed. Instead the manuscript jumps directly to modellization and leaves behind any thorough analysis of the collected data. (iii) I have serious concerns about the chosen experimental procedures for HEI10 intensity measurements (see below)

Major points:

- The image analysis procedure: Fluorescent image quantification is a difficult point and requires very rigorous experimental design to be achieved properly. Here, I cannot see that the authors paid attention to that but instead they chose to work after the acquisitions on the normalization of the fluorescent signals. This is even more of a problem here, since the authors use SIM microscopy which is based on structured illumination and image reconstruction. This technique is particularly prone to artifacts and introduces nonlinear modifications of the signals that, to my opinion, are hardly compatible with intensity quantification. If I have no doubt that their procedure allows the identification and mapping of HEI10 foci on SC, I'm not sure (to say the least) that intensity quantification is relevant. I think quantification of SIM images requires certain calibration of the experiments which do not appear anywhere here.

We appreciate the reviewer's concerns about intensity quantification, and possible issues due to normalization as well as nonlinear effects/artefacts potentially present in SIM microscopy. The first point to make is that intensity quantification only makes up a minority of our data (Figs. 1b,2kl,3d,4f in the main paper), most of which instead concerns positioning and number of HEI10 foci. Hence, all of our conclusions still stand even without intensity quantification. Nevertheless, we believe that our analysis of foci intensity is robust, as we now discuss.

The reviewer is right to point out the inherent problems with comparing absolute fluorescence intensity between cells. From our experience, fluorescence intensity is highly variable, both between slides and also between cells on the same slide (see below) due to differences in factors such as antibody binding and the local slide environment. This makes comparing absolute intensity values between cells unfeasible. However, normalizing HEI10 focus intensity values, relative to the total summed intensity of all HEI10 foci within the same cell (which will all have experienced similar imaging conditions) mitigates this intercellular variability. It is clear from the reviewer's comment that our reason for taking this approach was not sufficiently justified within the original manuscript and we have added additional text to the main text and methods section in order to clarify this point. (see lines 82-90 and lines 436-439 of the revised manuscript).

It is also true that 3D-SIM is prone to reconstruction artefacts. However, the introduction of artefacts is substantially reduced when imaging very thin samples (only a few microns thick) with very high signal contrast and that have been labelled with very photostable Alexa Fluor fluorophores (as we have done in this paper). Another advantage of comparing relative intensity values is that it allowed us to minimise the introduction of artefacts into our images by optimising the imaging conditions for each cell by adjusting the laser intensity and camera gain values. We have added to the methods section details of how the 3D-SIM channel-alignment was calibrated and how 3D-SIM imaging conditions were optimised. (see lines 370-389 of the revised manuscript).

However, we disagree with the statement that 3D-SIM introduces 'nonlinear modifications of the signal that...are hardly compatible with intensity measurements'. It is our understanding that 3D-SIM reconstructions (after estimating illumination parameters) use essentially linear processing and, therefore, retain relative intensity differences that are present within the raw image. To clarify this, we contacted the microscope manufacturer, Zeiss, who confirmed that the 3D SIM processing used on the Elyra microscope is derived from the algorithms devised by Gustafsson¹ which are linear. In addition, the suitability of SIM-imaging for intensity quantification has previously been assessed versus widefield epifluorescence and confocal laser scanning microscopy for the quantitative analysis of microtubule bundles and was actually found to be superior to both of these alternative imaging methods². We therefore have no reason to suspect that the conclusions we have drawn regarding HEI10 focal intensity are a product of our experimental methodology as opposed to a genuine biological phenomenon.

However, to further minimise any reservations the reviewer may have regarding our methodology and to lend additional weight to our conclusions, we have re-analysed HEI10 focal fluorescent intensity using an alternative image analysis approach (FociPicker3D³) in unreconstructed pseudo-widefield⁴ images. These were generated by simply averaging the images from the different illumination patterns and their associated SIM reconstructions in a sub-sample of ten wild-type late-pachytene images that were

used in this study. We have added an additional figure (Extended Data Fig. 2) and additional text (see lines 98-103 and 476-513 of the revised manuscript) to describe this approach. In summary, using this alternative method we identify the same negative correlation between late-HEI10 relative focus intensity and focus number per bivalent in both pseudo-widefield and 3D-SIM images, and we show that the relative maximum intensity values for the same foci in the different image types are linearly correlated. We hope that this additional analysis, combined with the literature confirming the linear nature of 3D-SIM image reconstruction, will be sufficient to reassure the reviewer about our conclusions. In particular, our fluorescent intensity quantifications of late-HEI10 foci are neither a product of the imaging approach (3D-SIM reconstruction) nor of the image-analysis methodology used in this study.

- Issues with the normalization procedures:
 1. They are particularly difficult to understand.

We have rewritten the methods section to clarify the normalization procedures used in the analysis.

2. It seems that there have been several layers of normalization of the HEI10 signal (each involving subtraction of "background", cut off etc... I'm not sure that, at the end of such treatment (several normalization rounds based on different parameters), that statistical analyses can be done.

As mentioned above, we have rewritten the methods section to more clearly describe the steps involved. There are two main tasks which include normalization: identification of the foci, and quantification of foci intensities. For foci identification, the HEI10 intensities for each trace were normalized to have zero mean and unit variance (a simple transformation commonly used as a first step in data analysis), before applying criteria to identify peaks, and then selecting the subsets of these peaks which were foci. For the quantification of foci intensities, there are two steps between the processed SIM images and the plotted intensities – subtraction of the uniform background fluorescence levels (median intensity of HEI10 on each trace), and per-cell normalization relative to the total intensities of the foci within the cell. Neither of these steps preclude the statistical analysis of the results. We also note in the revised manuscript (line 418) that we obtain equivalent results if, for foci identification, we subtract the median of each trace rather than the mean.

Whilst we do not see a problem with our approach, we believe that the additional analysis we have performed that utilises an alternative focus identification approach (FociPicker3D – see above) will allay the fears of the reviewer that our conclusions are a product of a normalisation bias within our image analysis pipeline. We have also updated the manuscript to include additional statistical analysis of the data obtained using our original image-analysis pipeline that demonstrates significant differences between focus intensities for single, double and triple foci (Fig. 1b).

3. From my point of view, it is highly questionable to normalize by HEI10 intensity. Above all considering that the whole point of the study is centered around modulation of HEI10 quantity along chromosomes...If the authors want to understand how HEI10 quantity is evolving, they should have normalized by another cellular component. DAPI intensity? ZYP1 signal? Axial element labeling?

I cannot understand why these analyses haven't been done. They would allow to compare HEI10 intensities between two cells even at different pachytene stages (early/middle/late) and validate or not that HEI10 intensity is constant from early to late pachytene, which if I understand properly is one of the founding hypotheses of their model.

In principle, this is an excellent suggestion, but we were unable to find a way to usefully and accurately normalise the HEI10 signal against a different protein, with a different localisation profile, that is labelled using a different primary and secondary antibody combination with a different fluorescent emission wavelength. As a result, the fluctuations in the signals of the HEI10 and the second standard would not necessarily be correlated, making it unreliable to systematically compare relative intensity differences. However, we were inspired by the reviewer's suggestion of harnessing the ZYP1 signal to validate our quantitative image analysis approach and confirm that there was no systematic bias within our imaging pipeline that would lead to different fluorescent intensity values being detected at different bivalent positions. We found that when we plotted the ZYP1 intensity along bivalents against bivalent position that there was very little fluctuation in average ZYP1 intensity within the data, consistent with there being no systematic bias within our image analysis methodology (Response Fig. 1).

As previously mentioned, due to the differences in absolute intensity between cells it is unfeasible to directly demonstrate that HEI10 concentration per bivalent is constant from early to late pachytene. As the reviewer rightly points out, this is indeed one of the founding hypotheses of the model. Furthermore, our rationale for taking a modelling approach was due to the difficulty of experimentally demonstrating this type of protein behaviour *in planta*.

Response Fig. 1: A. Examples of ZYP1 trace intensities (obtained using the same pipeline as Fig. 1A) from six bivalents selected at random from the dataset, showing substantial fluctuations. **B.** Distribution of ZYP1 trace intensities along WT bivalents. Each trace was divided into ten sections of equal length, and the mean ZYP1 intensity in each section calculated. Violin plots show the distributions of these mean intensities over WT bivalents, suggesting that any systematic differences in ZYP1 intensity between the centre and ends of bivalents is limited.

According to the Materials and Methods section, authors can stage cells from one anther and use the remaining 5 to target a specific pachytene stage. So, I suggest that they mix stages on slides to prevent slide variability and be able to compare HEI10 intensity on cells at different pachytene stages. Acquisition on

the same slide and with same exposure time will allow them to compare HEI10 intensity either with SIM or with epifluorescence.

In principle, this is again an excellent suggestion, but from our extensive experience imaging fluorescently labelled *Arabidopsis* meiocytes, fluorescence intensity is also highly variable between cells on the same slide (as well as between cells on different slides, as the reviewer mentions). To experimentally demonstrate that this is the case we imaged 68 pachytene cells labelled with ZYP1 that were present on the same slide using epifluorescence microscopy and exactly the same imaging conditions for each cell (20ms exposure time and 40 stacks with an interval size of 0.22 μm) (Response Fig. 2). We then used our imaging pipeline to calculate the mean absolute ZYP1 intensity within a small ($\sim 2 \mu\text{m}$) stretch of traced synaptonemal complex from each cell (note: limitations in the resolution of epifluorescence microscopy prevented us from segmenting and tracing the entire end-to-end lengths of bivalents within these images) (Response Fig. 2abcd). Using this technique, we found that there were up to 6-fold differences in mean absolute intensity of SC associated ZYP1 between different cells on the same slide (Response Fig. 2e). To confirm that this result was not a product of our analysis pipeline, we also used a different, more simplistic, approach to measure the differences in ZYP1 intensity between cells by plotting a line transect across a maximum intensity projection of each cell and obtaining the maximum intensity value from that line transect (Response Fig. 2fg). This second approach yielded broadly similar results, indicating that there were up to 6-fold differences in the maximum absolute intensity of the ZYP1 signal in pachytene cells from the same slide measured using epifluorescence microscopy and the exact same imaging conditions (Response Fig. 2h).

Response Fig. 2: Comparison of absolute ZYP1 intensity values of cells on the same slide imaged using exactly the same imaging conditions. a, a maximum intensity projection of a wild-type pachytene cell labelled for ZYP1 and imaged using epi-fluorescence microscopy. A small section of ZYP1 signal that was traced in 3-dimensions using the Simple Neurite Tracer (SNT) plugin to Fiji is highlighted in yellow. **b, c, d**, outputs of our image analysis pipeline on highlighted SNT path. **b**, maximum intensity projection of segmented section of ZYP1 signal. **c**, orientation of voxels within the 3D-path to give a defined start (green) and finish (red) orientation. **d**, mean absolute ZYP1 intensity for each voxel along the segmented region. **e**, mean absolute ZYP1 intensity values per voxel for each overall traced 3D-path (calculated by dividing the sum of the mean absolute ZYP1 intensity values for each voxel of the traced bivalent by the total number of voxels in the trace); plotted for 68 cells that were imaged on the same slide. **f**, the same maximum intensity projection as **a** but with a yellow line indicating the position of a line transect that spans the width of the cell. **g**, plot showing the absolute ZYP1 intensity values for the line transect of the maximum intensity projection. **h**, maximum absolute ZYP1 intensity values from line transects across maximum intensity projections of 68 cells that were imaged on the same slide. Scale bars = 5 μm .

Taken together, this indicates that the experiment suggested by the reviewer would be unlikely to yield the straightforward results suggested and supports our reasoning for comparing relative HEI10 intensities between cells rather than absolute intensity values. While we have presented these results in detail within this document as they lend vital support to our response, we do not feel these results are of sufficient relevance to the overall narrative of this paper to justify inclusion within the main manuscript. We have, however, added additional text to the manuscript indicating that large differences in absolute intensity are even found between cells on the same slide (see line 86 in the revised manuscript).

For me this is the only way to prove what is written line 130: "Overall, the initial amount of HEI10 in the model is consistent with a fixed amount per μm of SC, as observed cytologically (Fig. 1b)." How things have been done here, there is absolutely no way to compare early and late HEI10 intensity so this assumption is definitely not true.

As the reviewer rightly points out, there is no way to accurately compare early and late HEI10 intensity using our data. However, it was not our intention to imply that this is the case in the sentence that the reviewer has highlighted. Instead, we are highlighting that we observe a positive correlation between relative HEI10 intensity and relative SC length, as shown in Fig. 1b, with relatively longer SC lengths being occupied by a greater relative fraction of overall HEI10. Thus, in the initial conditions of our model we have loaded a uniform amount of HEI10 per micron of SC length (i.e., longer bivalents will load more HEI10). We have added additional text to this sentence to avoid any misinterpretation (see lines 161-162 of the revised manuscript). Unfortunately, as discussed above, it is unfeasible to empirically demonstrate that there is a fixed amount of HEI10 from early to late pachytene and, thus, this remains an untested assumption within the model.

4. Nowhere in the text is it explained how are treated the data once normalized. The reader gets the impression that all data extracted from different cells are pooled together. If it is the case it makes absolutely no sense to compare intensity. Therefore, I guess (literally, because I cannot find the information anywhere) that the HEI10 focus intensities given are corresponding to a ratio, related to total cell intensity. This has to be explained properly.

We apologise to the reviewer that our approach of using per-cell relative intensities was not made sufficiently clear within the original manuscript. We have now strengthened our explanations: for example, we have added additional text (lines 82-90) within the manuscript and the legend of Fig. 1 that makes it clearer that we are comparing per-cell relative intensity values, rather than absolute intensity values. Data is normalized for each cell before being pooled from multiple cells. With this additional text more clearly describing our justification for comparing per-cell relative intensity values, we believe that our approach is now much more clearly described.

-Data analyses: I don't understand why the collected data are so poorly analyzed. Hardly any quantitative analyze of the data is provided before the modelling part. What is the average number of HEI10 late foci per cell? Do these numbers fit with already known data?

To address this oversight, we have added additional text, the mean total late-HEI10 foci per cell values and mean total SC length per cell values, as obtained using our image analysis pipeline (see lines 74-79 and lines 237-239 of the revised manuscript).

The authors say that HEI10 foci intensity is stronger when HEI10 foci are single than when multiple. What is the range of intensity variation we are talking of? Is it true whatever is the total amount of HEI10 foci per cell?

As discussed above, these are per-cell relative intensity values and the range is now shown in Fig. 1b. The per-cell relative intensity data in this figure is collected from bivalents from all cells regardless of the total number of late-HEI10 foci per cell and, hence, we can conclude that relationship between focus intensity and focus number per bivalent is consistent regardless of the total number of late-HEI10 foci within a cell.

- Another important point is to understand how the authors cope with the timing of appearance of HEI10 late foci? Could the different of intensity they detect reflect a difference in timing? The authors treat their data of late HEI10 foci as if all the cells had reached a final state but instead it is very likely that their late pachytene cells contain an heterogenous population of cells. How far the differences of intensity they detect could reflect a different in the stage at which the cells are?

The reviewer is right to point out the likely heterogenous nature of the late-pachytene cells and that differences in the timing of progression through pachytene will have possible effects on the number of HEI10 foci observed within the cell (indeed, the effect of meiotic progression on HEI10 focus number is an integral part of our model). However, this potentially confounding effect of differences in timing in late-pachytene cells once again provides further justification for comparing per-cell relative intensity values. Any differences in intensity that we detect are, therefore, relative only to foci that are within the same cell and, thus, are at exactly the same pachytene stage.

Additionally, it is worth noting that our model predicts that fluctuations in final HEI10 focus numbers would be minimal within the last few hours of pachytene (Fig. 2j) and, hence, we would expect the small differences in meiotic progression that are inevitably a feature of our collected late-pachytene data would have little influence when we are comparing experimental data with model predictions (Fig. 2i).

- The negative correlation between HEI10 intensity and HEI10 foci number is a

very strong statement in this work. I think the authors absolutely need to strengthen it.

1) I suggest that they repeat HEI10 intensity measurement using epifluorescence microscopy imaging and proper image acquisition calibrations.

As noted above, we have repeated these measurements using an alternative image-analysis approach and using pseudo-widefield images (Extended Data Fig. 2). We believe this provides significant further evidence to support our conclusions indicating a negative correlation between relative HEI10 intensity and HEI10 focus number.

2) If the authors are right about the negative correlation between HEI10 focus intensity and number of foci per bivalent and that this difference is not explained by a staging issue, they should find the same correlation at later stages (diakinesis) when they are sure to look at mature HEI10 foci. This is a control relatively easy to achieve.

While this is an excellent suggestion, we think this control may be difficult to perform and, even if achievable, may not then be easy to interpret. Firstly, from a technical perspective, immunofluorescent labelling of HEI10 foci in *Arabidopsis* diakinesis cells requires a different slide preparation technique in which the material is fixed in 3:1 ethanol:acetic-acid, rather than paraformaldehyde. From the authors' own experience, this aggressive fixation method is quite damaging to proteins and their epitopes and may prevent the uniform and reliable binding of primary antibodies to the fixed material that would allow for accurate protein quantification based on immunofluorescent intensity. Secondly, it is not necessarily true that we would expect the same correlation in HEI10 focal intensity to be found post-pachytene (e.g., there may be additional post-patterning or post-crossover-designation effects on HEI10 localisation of which we are unaware) and an absence of intensity vs. number correlation at this later stage would not necessarily contradict the presence of such a correlation, as we have shown, at an earlier stage. Indeed, either of these two reasons may explain why this reasonably clear relationship between focal intensity and number has not previously been identified in *Arabidopsis*.

- Line 162/166: it is stated that concerning the case of double CO "the size of the leftmost of the late-HEI10 intensity peak was generally greater when the rightmost peak was positioned close to the right bivalent end" and that this result of the model was confirmed on the cytological data"... But I really cannot see how left and right end of a SC can be defined on bivalents that have no orientations!

The left/right definition of bivalent ends is entirely arbitrary and was chosen randomly for each bivalent: it makes no difference for this measure which end is chosen. We have changed the wording in the manuscript and figure axes to clarify this issue.

Minor points:

- I129: "initial amounts of HEI10 at foci near bivalent ends were greater than for those elsewhere (Fig. 2b)." I don't see anything like that on 2b.

This statement referred to the initial condition in the model, as implemented by the function f illustrated in Fig 2b. We have clarified the relevant sentence.

- I127: "To account for the preferential initiation of synapsis near telomeres"

Telomere driven synapsis evidence are very scarce in *A. thaliana* as far as I know

While it is true that interstitial synapsis initiation sites are present in mid-/late-zygotene *Arabidopsis* cells, it has previously been observed that synapsis begins at regions close to the telomere in early-zygotene. We have added an additional reference (Lopez *et al.*, 2008)⁵ to the text to further support this point.

- The authors should be careful not to use "CO" for "HEI10 foci". It's a shortcut that can very well turn wrong, above all when analyzing pachytene HEI10 foci

Thank you -- we have corrected the main text and figure legends to address this point.

Reviewer #2 (Remarks to the Author):

As Morgan *et al.* notice, CO patterning is poorly understood, particularly as it comes to elucidating CO interference. The authors propose a neat mathematical model that relies on cytological analysis of HEI10, a key protein involved in the progression of recombination intermediates to COs. Using this model, the authors can predict CO distribution in wild-type *Arabidopsis* as well as mutant lines with reduced and increased expression of HEI10. Overall, this is an interesting piece of work. There are a few areas where the manuscript needs improvement.

1. I worry that the underlying mechanistic explanation on HEI10 protein diffusion along chromosomes is very speculative. This process has not been experimentally observed and there may be many questions on how it could actually work in the context of chromosomes and chromatin. Is there any experimental evidence the authors could cite in support of this model?

We currently cite the only data that is (to our knowledge) available. The Stauffer *et al.*, paper⁶ uses FRAP and FCS data to demonstrate diffusion of the HEI10 function homologue, ZHP-3, along chromosomes in *C. elegans*. We point out the difficulties in experimentally observing this phenomenon in *Arabidopsis* in the text (lines 118-120) and, indeed, this is a primary justification for using a modelling approach.

2. Why would sites with more HEI10 protein be more likely to generate COs?

We thank the reviewer for raising this important question and apologise for not addressing this within the original manuscript. There are, in fact, many examples in the literature demonstrating that the dynamic clustering of signalling proteins into concentrated foci (e.g. the WNT signalosome) is required to increase their local concentration and signalling activity. For HEI10, this increased local concentration and, hence, increased signalling activity, could presumably be required to drive CO designation. We have added extra text and citations to the manuscript to clarify this point (see lines 110-113 of the revised manuscript)

3. The authors could consider examining HEI10 localization in a line with drastically reduced DSB numbers. If the presence of large vs small foci is dependent on the amount of HEI10 protein being produced, only large foci should be found.

While this would be an intriguing suggestion, it is also quite probable that lines with drastically reduced DSB numbers may also have other pleiotropic effects (e.g., on synapsis) that could also affect the dynamics of HEI10 loading and, hence, the results of such an experiment might be hard to interpret.

4. Is there evidence for CO numbers correlating with SC length in *Arabidopsis* other than the male-female relationship? As male-female recombination differences are not that well understood at the molecular level, it might be that SC length is not the only or not the most important difference. Can the authors examine the lines from Xue *et al.* to see whether they exhibit shorter SC?

While we are aware that other differences may (and, indeed, are likely to) exist between male and female meiosis, we demonstrate that in our model the SC length difference that has been published is sufficient to alter CO frequency and positioning. This is a prediction from the model that fits with all known experimental data on the subject. Unfortunately, we cannot fit or test or model against unknown features of male/female meiosis. However, the results of future studies into male/female differences in *Arabidopsis* meiosis will, as the reviewer suggests, offer excellent opportunities for further enhancing and optimising our model. Temperature is also known to affect SC length in *Arabidopsis*, but temperature would also be likely to affect protein diffusion/reaction kinetics and it would, therefore, be difficult to draw conclusions from experiments at variable temperatures. Finally, analysing the lines from Xue *et al.*, would require much additional experimental work, akin to the same volume of cytological work we have already performed in this study, and, while the results would be interesting, we do not believe they would qualitatively alter our conclusions. We think this might be a very worthwhile avenue for future research and we thank the reviewer for this suggestion.

5. Line 67-68: “widely varying HEI10 intensity as a function of position in μm along the SC” –is there a possibility that some of this intensity variation is caused by variable accessibility of HEI10 molecules to the antibody rather than just the HEI10 protein amount? Is there a way to normalize HEI10 fluorescence, for example by immuno-colocalizing with ASY1, H1 histone or anything else that should show uniform staining along chromosomes?

We thank the reviewer for this excellent suggestion of utilising an alternative immuno-fluorescently labelled protein to validate its uniform staining and, hence, antibody penetration, from end-to-end along individual bivalents. We found that when we plotted the mean absolute ZYP1 intensity along bivalents against bivalent position, using the same approach we had taken for HEI10, that there was very little fluctuation in average ZYP1 intensity within the data, consistent with there being no systematic bias in antibody accessibility or within our image analysis methodology (see Response Fig. 1).

6. Line 128: Does synapsis in *Arabidopsis* indeed start exclusively at telomeres? I thought there were also data indicating that synapsis initiation was interstitial.

While it is true that interstitial synapsis initiation sites are present in mid-/late-zygotene *Arabidopsis* cells, it has previously been observed that synapsis begins at regions close to the telomere in early-zygotene. We have added an additional reference (Lopez *et al.*, 2008)⁵ to the text to further support this point.

7. Lines 175-178: Could the authors elaborate on how exactly this simulation was done? Did it assume the same number of DSB/RI in male and female meiosis? Is this a fair assumption?

We state that the simulation was done by: “reducing SC length in our model (with otherwise default parameters)”. Therefore, RI density remained the same (which would mean that the number of RIs in females would reduce, as there is a density of 1 RI per 2um of SC). We now state this explicitly in the relevant text (line 212). As a difference in SC length is the only parameter in our model that has been experimentally demonstrated to change between male and female meiocytes (based on available published data), it seemed reasonable to only change this parameter – particularly as this change was solely sufficient to recapitulate the experimental CO patterning data.

8. Lines 149-152: Could the authors elaborate on the added benefit of their analysis in this case. Xue et al. already explained the extent of CO reduction by just considering the need for an obligate CO and the correlation between COs and DSBs.

The section of text highlighted by the reviewer indicates that the model fits, and is in agreement with, the CO homeostasis data from the paper by Xue *et al.*. The Xue *et al.*, paper provides an excellent experimental demonstration of the effect of reduced DSB number on CO frequency in *Arabidopsis*. Fitting our model to this data is an important aspect of the model validation process, enabling us to demonstrate that our model fits well with known *Arabidopsis* recombination data and, therefore, can offer a plausible explanation for the mechanism underlying CO positioning in *Arabidopsis*. To be clear, we have not redone the experimental work or performed any additional analysis of the work done by Xue et al., but have merely fitted our model to it. However, we have added additional text to the manuscript that further highlights the insightful conclusions from Xue *et al.*, regarding CO homeostasis and the obligate CO (see lines 183-184 of the revised manuscript)

9. Lines 183-185 and on: Rather than speculating, the authors could determine this pattern using biological data, at least for the pericentromeric regions. This can be done, for example, by colocalizing centromere proteins with either Hei10 or DMC1/RAD51.

While the experiment this reviewer suggests is, in theory, possible, it would require a substantial amount of work and would, in practice, be even more complex than the cytological mapping of crossover sites that we have performed in this study (which, we note, is still technically very challenging and, hence, we are the first group to have achieved this feat in *Arabidopsis*). Additionally, in the development of this paper we did try immunolocalization against CENH3 in *Arabidopsis* but, unfortunately, we could not get the antibody to work sufficiently well in our material. The segmentation of chromosomes at an earlier stage of meiosis (leptotene), when DMC1 and RAD51 foci are present, would also be very complex and possibly not achievable due to the especially tangled nature of *Arabidopsis* chromosomes at this early stage. Taken together, while this is a great suggestion, from our experience we do not feel that it is yet technically feasible to carry out cytological mapping of DSB sites in *Arabidopsis*.

Minor issues:

1. Line 25: I would suggest “crossover positions can be identified” rather than “explained”

We feel that using the word ‘identified’ in this context implies that the coarsening model is capable of experimentally determining the positions of crossover, which is not the case. The model predicts the statistical distribution of crossover positions, based on a particular set of parameters, which can then be tested against the crossover position distributions that are ‘identified’ using our cytological mapping approach. We therefore feel the term ‘explained’ is more appropriate and is less likely to lead to confusion.

2. Line 85: “SC length dependent manner”: this phrase is not clear. Could the authors elaborate?

We have added additional text to clarify this point (see lines 104-105 of the revised manuscript).

Reviewer #3 (Remarks to the Author):

The manuscript entitled “Diffusion-mediated HEI10 coarsening 1 regulates meiotic crossover positioning” by Chris Morgan and colleagues addresses the mechanism responsible for the crossover between sister chromosomes observed during meiosis and in particular why all chromosomes must receive at least one “obligatory crossover” while crossovers are prevented from occurring near one another by “crossover interference”. Although this phenomenon is known for more than 100 years, the reasons behind it are still unclear and in this study, the authors investigate this problem by using quantitative super-resolution cytogenetics and mathematical modelling in *Arabidopsis thaliana*.

The authors showed that crossover positions could be explained by a predictive, diffusion-mediated coarsening model, in which large, approximately evenly-spaced HEI10 foci grow at the expense of smaller, closely-spaced clusters. They interpret crossover interference observed in *Arabidopsis* in terms of the coarsening process of the model. Their model recapitulated the “gain of function” and “loss of function” phenotypes observed in the *Arabidopsis thaliana* wild-type with the over-expressor of the conserved E3 ligase HEI10 and a *hei10* heterozygous *line5* model, respectively, by changing one single model parameter, the HEI10 dosage.

Crossover between homologous chromosomes is of fundamental importance to create genetic diversity within offspring. This study puts forward a plausible mechanism implemented in simple and yet elegant mathematical model which is quantitatively in agreement with the presented super-resolution cytogenetics data. The manuscript is concisely and nicely written. That being said, I found the following points important to address.

Majors:

1. The authors investigated this problem in *Arabidopsis thaliana*, although as their findings might be extrapolated to other eukaryotes (as they indeed speculated), the manuscript title should reflect the name of this organism.

The reviewer is right to point this out. We have amended the title to address both this point and the point below. New title: 'Diffusion-mediated HEI10 coarsening can explain meiotic crossover positioning in Arabidopsis'.

2. The authors did not demonstrate that diffusion is the mechanism controlling HEI10 dynamics, which in turn would regulate meiotic crossover positioning. Instead, they conceived a model assuming diffusion of HEI10, which nicely recapitulates their experimental results. Hence, the agreement between the model and the experimental data suggests that spreading of HEI10 could be in fact driven by diffusion. Nevertheless, other not yet tested mechanisms could also be at work. Thus, as tempting as it might be to invoke this mechanism, I believe that some caution could be exercised in the article's title.

See our response to point 1.

3. The authors find that HEI10 levels correlate with SC lengths. Nevertheless, this correlation is not obvious by visual inspection of the plots (Fig. 1 b) and a measure of this linear correlation is missing.

Thank you - we have added the R-squared coefficient to the plots as a measure of this linear correlation.

4. Regarding the initial condition for the SC lengths, some effort could be done to better explain how these lengths are determined.

a) In the Extended Data Fig. 5, the authors showed a table of SC lengths used for simulations. It is not clear to me what is the origin of these data. Are these values measured by the authors? If so, they should stay so, if not, a proper reference should be added.

We obtained these SC lengths from the experimentally traced SC lengths. This has been clarified in the caption to Extended Data Fig 6, and noted in the Methods section (lines 521-528). We have made a small change to the values used for these parameters (in Extended Data Fig. 6), as we realised that the previous values were based on an incomplete subset of the data. This small change in parameter values makes only a very small difference to the simulation results.

b) Between lines 101-104 of the main text, the authors wrote "a synapsed pachytene bivalent is considered as a one-dimensional structure, with the simulated SC length drawn at random from the experimental bivalent SC length distributions". Then, in the Methods section "Mathematical modelling" bellow line 361, the authors stay "where the initial conditions incorporate stochasticity [...] in the length of the SC" and bellow, in the same section, "for each of the five SC lengths, a length L was generated at random from the appropriate empirical distribution, approximated by a normal distribution (with values clamped to the range within three standard deviations of the mean)". If I understood correctly, the authors modeled the 5 SC lengths (from the 5 chromosomes) with 5 normal distributions, each of them parametrized with the mu and sigma extracted from the table?

Yes, the reviewer understood correctly. We have added additional text in the Methods section to explain how these distributions were obtained from the experimental data, and how they were used in the simulations.

5. How exactly the initial conditions for RI were implemented? The authors wrote (main text, lines 104-106) “RIs were placed at random positions along the five bivalents, in accordance with the cytologically defined random or weakly-interfering nature of early recombination nodules in plants”. In lines 373-377, methods section, they wrote “RIs were then placed at random positions x_1, x_2, \dots, x_N , along the SC, where x denotes distance along the SC. For each SC with randomly-generated length, the number of RIs was chosen to give 1 RI per $2 \mu\text{m}$, rounded to the nearest integer”. What is the statistical distribution of the RI?

We have now made the distribution (sampled from a uniform distribution) explicit in the Methods section. We have also changed the description of the Methods section to make clear that, first, the length of the bivalent is generated, then the number of RIs on each bivalent is determined, and finally these RIs are placed randomly (with uniform distribution) along the SCs.

6. The authors consider that HEI10 escapes from the immobile foci back onto the SC at a rate β , which they assume is the popular Hill-expression. Regardless the fact that maybe more effort could be made to make clear why this expression was chosen, an important question is how robust is the model to this assumption. What would happen if other expressions were tested?

To address this, we have explored simulations of the model with an alternative form for the escape rate beta, namely $\beta(C) = \beta_c / (1 + \delta e^{C/K_c})$, and found (Extended Data Fig. 7) that, with suitably chosen parameter values, we could obtain broadly equivalent results for the wild type. We have added some discussion (lines 594-600) about the necessary features of this escape rate for the model to give sensible outcomes.

7. The authors wrote (Lines 193-196) “After fitting and validating model parameters using wild-type data, we next tested how cytological data obtained from lines exhibiting variable HEI10 dosage compared with model predictions. For these cases, all earlier parameter values were fixed except the HEI10 concentration, which was treated as a single fitting parameter”. First, I guess the authors are referring to the initial concentration of HEI10 (c_0 in the model), so it would be better if the authors would explicitly state that. Second, and more important, by considering c_0 as the only fitting parameters, the authors are assuming that the experimental conditions behind HEI10 over-expressor line C2 and the *hei10-2* +/- heterozygous T-DNA insertion line would be equivalent to the wild-type with the only exception of this parameter. Although this assumption is certainly straightforward and reasonable, I am wondering whether, for instance, the parameters reflecting the initial RI loading in these two experimental conditions could be different from the wild type condition as well. Why the authors discarded this possibility and others?

We have amended the text to make it clear we are referring to c_0 , C_0 and σ (see line 232 of the revised manuscript), changing both the HEI10 concentration on the SCs and the loading of HEI10 at the RIs by the same multiplicative factor; this was chosen as the simplest assumption for the effect of changes in HEI10 levels. Regarding the second point, it intuitively makes more sense to start by changing the initial HEI10 levels in our

model, as these parameters are experimentally known to vary between the lines. As changing the initial HEI10 levels in the model was found to be sufficient to recapitulate the experimental data, it seemed unnecessary to make any assumptions regarding variation in other model parameters. This is particularly the case as we do not expect HEI10 expression levels to affect the diffusion of HEI10 along SCs, or to (directly) affect the dynamics of HEI10 at RIs.

8. (Lines 208-209) The fact that the bivalent CO number was zero (Fig. 2 c right, Fig. 3 b right, Fig. 4 b right) is an emergent result of the simulations or a constrain imposed by the authors. If it is the first one, it is remarkable and should be made explicit and if it is the second, it should be made explicit.

We have added additional text in the main paper (lines 140-142) and Methods section of the manuscript (lines 602-610) discussing how the model avoids bivalents with zero COs under sufficiently large initial HEI10 loading. This feature is a natural consequence of the coarsening dynamics: provided HEI10 foci over a critical threshold form, the coarsening dynamics will take over, reducing the number of foci. However, the focus number cannot drop below 1, as after coarsening into a single focus, that focus is necessarily stable, as there is then nowhere else for HEI10 to accumulate.

9. The mathematical model here proposed by the authors has some resemblance with one we reported two years ago. In that model, a molecular motor (SpollIE) is able to bound/unbound to a DNA molecule, undergoes 1D-diffusion along the DNA molecule and changes their diffusion coefficient and binding/unbinding rates in a “special region” of the DNA molecule (SRS sequence) (Chara et al., 2018. Sequence-dependent catalytic regulation of the SpollIE motor activity ensures directionality of DNA translocation. *Sci Rep.* 8, 5254. <https://doi.org/10.1038/s41598-018-23400-8>). Although the here proposed model involves more than one “special regions” (RIs in this model) and, all in all, is addressing a different biological phenomenon, acknowledging the models resemblance could strength the current model plausibility since this modelling strategy proved to be successful in other scenario.

Thank you - we have mentioned this work in the manuscript (lines 142-145).

Minors:

In Fig. 1 b and its legend, the concentration of HEI10 along the SC (c) is not explicitly defined.

We assume the reviewer meant Fig 2b? We have edited the figure legend to define both c and C.

Legend Fig. 2 i. In the figure, we see red crosses and in the legend are referred as red circles.

Good spot! We have amended this.

The authors stay “We emphasize that the model is simple, with only five dynamical parameters and another five specifying initial conditions (parameter values in Fig. 2b and Methods)”. First, I might be wrong but I count in Fig. 2b 6 parameters corresponding to the initial conditions, and second, although I

understand the point, I think it is at least debatable to characterize an 11-parameters model as "simple".

We thank the reviewer for noticing our mistake. It is possible to eliminate a parameter (as the concentration units used in the model are arbitrary), but we have not done so for simplicity of presentation. We have revised this sentence to accommodate the reviewer's concerns, and also to note that there are other parameters (e.g., SC lengths) which are determined by the experimental data.

References:

1. Gustafsson, M. G. L. *et al.* Three-Dimensional Resolution Doubling in Wide-Field Fluorescence Microscopy by Structured Illumination. *Biophys. J.* **94**, 4957–4970 (2008).
2. Komis, G. *et al.* Dynamics and organization of cortical microtubules as revealed by superresolution structured illumination microscopy. *Plant Physiol.* **165**, 129–148 (2014).
3. Du, G. *et al.* Spatial dynamics of DNA damage response protein foci along the ion trajectory of high-LET particles. *Radiat. Res.* **176**, 706–715 (2011).
4. Ball, G. *et al.* SIMcheck: a Toolbox for Successful Super-resolution Structured Illumination Microscopy. *Sci. Rep.* **5**, 15915 (2015).
5. López, E., Pradillo, M., Romero, C., Santos, J. L. & Cuñado, N. Pairing and synapsis in wild type *Arabidopsis thaliana*. *Chromosom. Res.* **16**, 701–708 (2008).
6. Stauffer, W. T., Zhang, L. & Dernburg, A. Diffusion through a liquid crystalline compartment regulates meiotic recombination. in *Proc.SPIE* vol. 10888 (2019).

REVIEWER COMMENTS

Reviewer #2 (Remarks to the Author):

I was asked to comment on how the authors dealt with the suggestions from my review (#2) as well as those of Reviewer 1, who was not available for re-review.

Overall, the authors made a number of changes in the revised manuscript to include more information and improve clarity. However, they have not provided new experimental evidence, which I believe is needed to demonstrate biological relevance and validity of several assumptions of the model.

1. It is not clear to me why the authors have not at least tried normalising the HEI10 intensity data using a fluorescently-labelled antibody to another protein. It is true that some factors affecting signal intensity, such as antibody affinity to its antigen target would be different. However, there are several factors, such as cell-to-cell variation in fixation efficiency, antibody penetration, and imaging exposure conditions that should be similar for both antibodies. I am not convinced that signal intensity fluctuations between HEI10 and an antibody used as a normalisation standard would not be correlated. The analysis presented in Resp. Figure 1A is not terribly persuasive as a substitute for normalising against another protein-antibody pair. How were the six chromosomes selected? Where they on different slides?

2. I do not understand why examining HEI10 foci in diakinesis would require a slide preparation method that is incompatible with successful antibody labelling. There have been a number of antibody localisation studies in Arabidopsis cell fixed using the ethanol:acetic-acid fixative. I second reviewer 1 in believing that examining diplotene or diakinesis bivalents may provide a corroboration of the assumption of a negative correlation between HEI10 focus number and intensity.

3. I am still not convinced of the biological basis and significance of the model assuming HEI10 diffusion and its role in inducing CO formation depending on local protein concentration. The evidence provided by the authors for anything like that indeed existing is indirect and sparse. Without further experimental data, the model can only be treated as a speculation.

4. I also have continued reservations about other assumptions of the model, including the relationship between SC length and CO number. Other than the male-female relationship, I am not aware of any other data that would support this assumption in Arabidopsis. Furthermore, recent studies show abundant CO formation in Arabidopsis in the absence of ZYP1, so it is questionable at this point whether an SC length - CO number correlation exists in this species at all. I disagree that analysing SC length in Xue et al. lines would be such a substantial amount of work. The assumption of an SC length - CO number correlation needs experimental validation or it should be removed from the manuscript.

5. I disagree that it is not possible to identify HEI10/DMC1/RAD51 foci overlapping the relatively small pericentromeric region of Arabidopsis chromosomes. If the CENH3 antibody does not work well in authors' hands, immuno-FISH protocols are well established in Arabidopsis. Furthermore, DMC1/RAD51 are abundant on Arabidopsis chromosomes during zygotene – no need to examine leptotene cells.

Reviewer #3 (Remarks to the Author):

The authors have addressed all the points that I raised. The article constitutes the original and sound answer to an interesting problem over 100 years old. In my opinion, the article is ready for publication.

Prof. Dr. Osvaldo Chara

Dear Chuanfu,

Thank you for again forwarding the reviewer comments. We have addressed all the new points raised by Reviewer #2 (see below), including additional experimental data and analysis. We therefore hope that the manuscript will now be suitable for publication in Nature Communications.

Kind regards,
Martin Howard.

Reviewer #2

1. It is not clear to me why the authors have not at least tried normalising the HEI10 intensity data using a fluorescently-labelled antibody to another protein. It is true that some factors affecting signal intensity, such as antibody affinity to its antigen target would be different. However, there are several factors, such as cell-to-cell variation in fixation efficiency, antibody penetration, and imaging exposure conditions that should be similar for both antibodies. I am not convinced that signal intensity fluctuations between HEI10 and an antibody used as a normalisation standard would not be correlated. The analysis presented in Resp. Figure 1A is not terribly persuasive as a substitute for normalising against another protein-antibody pair. How were the six chromosomes selected? Where they on different slides?

It was not possible to normalise HEI10 intensity in our original dataset using the suggested approach as microscope acquisition parameters for every colour channel were optimised for each image to limit the introduction of SIM reconstruction artefacts (see previous response). To test the feasibility of the approach suggested by the reviewers we acquired an additional 3D-SIM imaging dataset (a mix of 27 early/mid/late pachytene cells from 3 wild-type plants) labelled with DAPI, HEI10, ZYP1 and the cohesin subunit SMC3 (Response Fig. 3A). During this new round of imaging, the acquisition parameters (exposure time, laser power and gain level) were kept constant for each individual colour channel. The absolute fluorescence intensity along each traced bivalent (obtained using our image analysis pipeline, Response Fig. 3B) in each cell, minus the median intensity value of the whole image to remove background, was summed and divided by the bivalent length (in pixels) for each colour channel to give a 'mean bivalent associated fluorescence intensity' value for each protein, and DAPI, for each bivalent. As shown in Response Fig. 3C, there was substantial cell-to-cell variation in mean bivalent associated fluorescence intensity for each protein and DAPI, consistent with Response Fig. 2 (see our previous response). From this approach it does appear there is potentially a ~40% reduction in HEI10 intensity going from early to late pachytene, although there are large cell-to-cell variations (Response Fig. 3C, left panel). However, this apparent reduction is difficult to interpret: it may, in part or even wholly, reflect limited antibody access to larger late-HEI10 foci, where antibodies may only bind the top layer of protein. For example, such antibody accessibility issues are known to cause an underestimation of strong bands in quantitative western blot densitometry and this is, in part, why these experiments require careful calibration and standardisation (Pillai-Kastoori, Schutz-Geschwender and Harford, 2020). Furthermore, this data is still consistent with late-HEI10 foci containing a greater abundance of HEI10 molecules than early-foci because, if focus-associated HEI10 levels remained constant from early to late-pachytene, we would expect a tenfold reduction in mean HEI10 intensity concurrent with the reported tenfold reduction in foci number from early- to late-pachytene; which is not observed. Similar antibody accessibility issues could, in principle, influence the HEI10 foci relative intensity

measurements in the manuscript. However, using a per cell relative approach, we are only comparing HEI10 intensity within individual foci of the same stage (i.e., late-foci vs. late-foci). Hence, accessibility changes will be significantly reduced than when comparing early vs. late mean bivalent associated intensity, as in Response Fig. 3, where relative differences in focus size (and, hence, antibody accessibility) will be much greater.

Additionally, we did not find that the mean bivalent associated HEI10 intensity strongly correlated with the mean bivalent associated intensity of any of the other 3 fluorescence markers (Response Fig. 3D). Nor did we identify correlations in fluorescence intensity between any of the other markers, which were all chosen as their bivalent associated amount should remain roughly constant both between cells and from early to late pachytene (Response Fig. 3D). Hence, we found that normalising the mean bivalent associated HEI10 fluorescence intensity by dividing it by the mean bivalent associated fluorescence intensity of any of the other three fluorescence labels in the same cell did not significantly alter our results (Response Fig. 3E). Given the low confidence we have in the reliability and/or interpretability of this approach and analysis, for reasons mentioned above, we have decided not to include this data in the main manuscript.

With regard to Resp. Fig. 1A, the six bivalents were randomly selected (as previously mentioned) and all come from different cells. Two of the bivalents were from separate cells from the same plant on the same slide, whilst the remaining four bivalents were all from separate plants on separate slides.

2. I do not understand why examining HEI10 foci in diakinesis would require a slide preparation method that is incompatible with successful antibody labelling. There have been a number of antibody localisation studies in Arabidopsis cell fixed using the ethanol:acetic-acid fixative. I second reviewer 1 in believing that examining diplotene or diakinesis bivalents may provide a corroboration of the assumption of a negative correlation between HEI10 focus number and intensity.

It is important to point out that the negative correlation between HEI10 focus number and relative intensity is not a model assumption, but has been experimentally demonstrated in this study using pachytene 3D-SIM and pseudo-widefield imaging data, with two separate analysis methods (see Fig. 1b right hand panel and Extended Data Fig. 2e,f). However, to determine if this correlation persists in diakinesis cells we attempted to label ethanol:acetic-acid fixed cells using the published protocol (Chelysheva *et al.*, 2010). Unfortunately, whilst we found this protocol worked well with certain antibodies (e.g. for ASY1, Response Fig. 4), we could not obtain reliable staining of HEI10 foci in diakinesis cells that would facilitate quantitative immunofluorescent intensity analysis, despite trying two separate HEI10 antibodies (raised in rat or rabbit). This is consistent with the observations of Chelysheva *et al.*, 2010, that some antibodies are unstable under these preparation conditions (e.g. for RAD51 and DMC1 in the hands of Chelysheva *et al.*) as well as our own experience (see previous response). However, we reiterate the point from our previous response that even if a HEI10 focus number vs. intensity negative correlation was absent during diakinesis, this would not contradict the presence of this negative correlation (as we have experimentally demonstrated using multiple approaches) during the earlier pachytene stage of meiosis, when we propose the HEI10 coarsening dynamics takes place.

"[Redacted]"

Response Fig. 4. Immunolocalisation using ethanol:acetic-acid fixed meiocytes. Top row, a leptotene cell labelled with DAPI and ASY1 antibody. Middle row, a diakinesis cell labelled with DAPI and HEI10 antibody (raised in rat). Bottom row, a diakinesis cell labelled with DAPI and HEI10 antibody (raised in rabbit). Scale bars = 5 μ m.

3. I am still not convinced of the biological basis and significance of the model assuming HEI10 diffusion and its role in inducing CO formation depending on local protein concentration. The evidence provided by the authors for anything like that indeed existing is indirect and sparse. Without further experimental data, the model can only be treated as a speculation.

This manuscript combines both experimental observations and mathematical modelling. Theoretical studies can sometimes lack direct evidence for all of their underlying assumptions. Their usefulness then rests on whether they can explain existing data in a simple way and whether they make useful predictions that can be experimentally tested. The simple model we present here satisfies both these criteria: a large body of experimental data is reproduced with high precision, and we then verify experimentally model predictions from over/under expression of HEI10. Furthermore, the model presents a decisive shift in thinking for how COs are formed towards a diffusive coarsening model, where we emphasise that the available evidence (from *C. elegans*) does directly support HEI10 diffusion along the SC. Finally, many other researchers are currently examining the behaviour of HEI10 homologues in other organisms, potentially with live-imaging. We hope that our model will help to stimulate and focus these new experimental approaches.

4. I also have continued reservations about other assumptions of the model, including the relationship between SC length and CO number. Other than the male-female relationship, I am not aware of any other data that would support this assumption in *Arabidopsis*. Furthermore, recent studies show abundant CO formation in *Arabidopsis* in the absence of ZYP1, so it is questionable at this point whether an SC length - CO number correlation exists in this species at all. I disagree that analysing SC length in Xue *et al.* lines would be such a substantial amount of work. The assumption of an SC length - CO number correlation needs experimental validation or it should be removed from the manuscript.

The positive correlation between SC length and CO number has been demonstrated in a wide variety of organisms (see Kleckner, Storlazzi and Zickler, 2003 and references therein). Furthermore, we have now included in the manuscript the requested validation of this relationship from our own data in *Arabidopsis* (see lines 218-221 and new Extended Data Fig. 5). Hence, this relationship is certainly not a model assumption, but rather a model output that is now experimentally verified. The recent study in *Arabidopsis* that the reviewer refers to demonstrates that heterochiasmy is abolished in SC mutants (Capilla-Pérez *et al.*, 2021), with the authors concluding that male/female differences in CO rate are imposed by the SC. This points to a direct role for the SC (which differs in length between male and female *Arabidopsis* meiocytes) in mediating sex specific CO numbers. We have added additional text to the manuscript that references this new study and how it supports our model (see lines 153-157 and lines 221-223). However, to be clear, we do not propose that SC length is the only parameter that can impact CO number and in our CO homeostasis simulations (fitted using the data from Xue *et al.*) we did not alter SC length but only altered RI density (this has now been clarified in the text, lines 188-189). We would therefore not expect a difference in SC length in the lines used by Xue *et al.*, which is consistent with findings from maize in which lines with different DSB numbers display equivalent SC lengths (Sidhu *et al.*, 2015). For these reasons, and as we have now shown the requested validation of the SC length vs CO number correlation, we have not further analysed the Xue *et al.* data.

5. I disagree that it is not possible to identify HEI10/DMC1/RAD51 foci overlapping the relatively small pericentromeric region of *Arabidopsis* chromosomes. If the CENH3 antibody does not work well in authors' hands, immuno-FISH protocols are well established in *Arabidopsis*. Furthermore, DMC1/RAD51 are abundant on *Arabidopsis* chromosomes during zygotene – no need to examine leptotene cells.

As this technically challenging experiment is requested to support a relatively minor point in the paper, we have decided to remove the section on modelling the effects of the centromere and the associated supplementary figure from the manuscript.

References

- Capilla-Pérez, L. *et al.* (2021) 'The synaptonemal complex imposes crossover interference and heterochiasmy in *Arabidopsis*', *Proceedings of the National Academy of Sciences*, 118(12), p. e2023613118. doi: 10.1073/pnas.2023613118.
- Chelysheva, L. *et al.* (2010) 'An Easy Protocol for Studying Chromatin and Recombination Protein Dynamics during *Arabidopsis thaliana* Meiosis: Immunodetection of Cohesins, Histones and MLH1', *Cytogenetic and Genome Research*, 129(1–3), pp. 143–153. doi: 10.1159/000314096.
- Kleckner, N., Storlazzi, A. and Zickler, D. (2003) 'Coordinate variation in meiotic pachytene SC

length and total crossover/chiasma frequency under conditions of constant DNA length', *Trends in Genetics*. Elsevier, 19(11), pp. 623–628. doi: 10.1016/j.tig.2003.09.004.

Pillai-Kastoori, L., Schutz-Geschwender, A. R. and Harford, J. A. (2020) 'A systematic approach to quantitative Western blot analysis', *Analytical Biochemistry*, 593, p. 113608. doi: <https://doi.org/10.1016/j.ab.2020.113608>.

Sidhu, G. K. *et al.* (2015) 'Recombination patterns in maize reveal limits to crossover homeostasis', *Proceedings of the National Academy of Sciences*. National Academy of Sciences, 112(52), pp. 15982–15987. doi: 10.1073/pnas.1514265112.

REVIEWERS' COMMENTS

Reviewer #2 (Remarks to the Author):

1. I do appreciate that the authors have made a serious effort to address previous comments, including performing new experiments. I still feel, however, that the overall hypothesis that diffusion mediated changes in HEI10 foci size are the driving force behind CO patterning is tenuous given the paucity of direct experimental evidence for the actual existence of this mechanism. I agree with the authors that their computer simulation indicates a correlation. To avoid overstating the biological implications of the simulation conclusions, there should be a stronger distinction between the actual results and their potential interpretation. This is particularly important given that authors could not get to work the experiments that I have hoped would have supported the proposed mechanistic explanations, such as HEI10 immunostaining in late prophase samples.
2. I am confused by the authors' use of the fact that longer chromosomes have more COs as evidence for a relationship between CO number and SC length. The latter would have been the case if lines or meiocytes with more COs exhibited longer SC but such data do not exist in Arabidopsis. This issue requires clarification.
3. Lines 202 and 206 should be modified to indicate the tentative nature of these statements. These could be potential explanations but they are by no means certain.
4. Sentence in lines 153-157 seems to be overstating the results of the two recent *zyp1* mutant analyses. These papers indicate that *zyp1* is required for CO interference and assurance as well as HEI10 patterning. That this could be evidence in support of ZYP1 promoting cis-coarsening is a speculation of the authors of the current manuscript.

Dear Chuanfu,

Thank you for again forwarding the reviewer comments. We have addressed all the editorial requests and the new points raised by Reviewer #2 (see below). We therefore hope that the manuscript will now be suitable for publication in Nature Communications.

Kind regards,
Martin Howard

Reviewer #2 (Remarks to the Author):

1. I do appreciate that the authors have made a serious effort to address previous comments, including performing new experiments. I still feel, however, that the overall hypothesis that diffusion mediated changes in HEI10 foci size are the driving force behind CO patterning is tenuous given the paucity of direct experimental evidence for the actual existence of this mechanism. I agree with the authors that their computer simulation indicates a correlation. To avoid overstating the biological implications of the simulation conclusions, there should be a stronger distinction between the actual results and their potential interpretation. This is particularly important given that authors could not get to work the experiments that I have hoped would have supported the proposed mechanistic explanations, such as HEI10 immunostaining in late prophase samples.

To address this comment (and in line with the editorial requests) we have now edited the manuscript to include separate Results (with subheadings) and Discussion sections, thus providing a greater distinction between the experimental results, modelling results and biological interpretations. Specifically, with regard to the interpretations, we have moved the sections discussing other biological mechanisms (i.e., the bacterial SpoIII^E and Wnt-signalling mechanisms) and the recent work in *Arabidopsis zyp1* mutants in the context of our coarsening model to the Discussion section.

2. I am confused by the authors' use of the fact that longer chromosomes have more COs as evidence for a relationship between CO number and SC length. The latter would have been the case if lines or meiocytes with more COs exhibited longer SC but such data do not exist in *Arabidopsis*. This issue requires clarification.

We have added additional text to this section (see line 310 with tracked changes, line 222 of 'no markup' version) to clarify that bivalent length within our experimental data was measured in microns of SC length (rather than, for instance, megabases of DNA) and, hence, we demonstrate a tendency for bivalents with longer SC length to have more COs, both within the experimental dataset and model simulations (Supplementary Fig. 5).

3. Lines 202 and 206 should be modified to indicate the tentative nature of these statements. These could be potential explanations but they are by no means certain.

We have added additional text (see lines 289 and 293 with tracked changes, lines 204 and 208 of 'no markup' version) to make it clear that these statements specifically refer to the model, rather than the experimental data. Within the constraints of the mathematical model, we can be confident that the explanation we have set out is correct.

4. Sentence in lines 153-157 seems to be overstating the results of the two recent *zyp1* mutant analyses. These papers indicate that *zyp1* is required for CO interference and assurance as well as HEI10 patterning. That this could be evidence in support of ZYP1 promoting cis-coarsening is a speculation of the authors of the current manuscript.

We have re-worded this section and moved it to the discussion to make it clear that the role of the SC in cis-coarsening is our own independent proposal, based on the synthesis of results from this study and recent findings from other groups.